# Diffusive Operator Spreading for Random Unitary Free Fermion Circuits

Beatriz C. Dias[1,2], Masudul Haque[2,3,4] Pedro Ribeiro[1,5] and Paul A. McClarty [2]

**1** CeFEMA, Instituto Superior Técnico, Universidade de Lisboa, Av. Rovisco Pais,
1049-001 Lisboa, Portugal
**2** Max Planck Institute for the Physics of Complex Systems, Nöthnitzer Str. 38, 01187
Dresden, Germany
**3** Institut für Theoretische Physik, Technische Universität Dresden, 01062 Dresden,
Germany
**4** Department of Theoretical Physics, Maynooth University, Co. Kildare, Ireland
**5** Beijing Computational Science Research Center, Beijing 100084, China
⋆ CorrespondingAuthor@email.address

May 11, 2022

## Abstract

**We study a model of free fermions on a chain with dynamics generated by random unitary gates acting on nearest neighbor bonds and present an exact calculation of out-of-time-ordered correlators. We consider three distinct cases: the random circuit with spatio-temporal disorder (i) with and (ii) without particle number conservation and (iii) the particle non-conserving case with purely temporal disorder. In all three cases, temporal disorder causes diffusive operator spreading and $\sim \sqrt{t}$ entanglement growth. We show that operator scrambling is strongly constrained in these random unitary circuits. The behavior of these models lies in sharp contrast to Anderson localization for the case of static disorder, and to the ballistic behavior and efficient scrambling observed both in evolution under interacting clean Hamiltonians and in fully random unitary quantum circuits.**

# 1 Introduction

Understanding quantum dynamics in many-body systems far from equilibrium is a central issue in contemporary physics. In the spirit of finding universal features in many-body dynamics, random unitary circuits [1–8] have been intensively studied in the last few years. In such models, instead of smooth temporal evolution, time is discrete and local random unitary gates are applied to some underlying degrees of freedom. Since there is no Hamiltonian dynamics, energy conservation is sacrificed in order to uncover generic features of local dynamics. These models capture the scrambling and spreading of quantum information including the initial linear growth of bipartite entanglement from an initial product state to a fully random state with Page entanglement [9]. An observable that is particularly suited to quantifying information spreading is the degree to which spatially separated local operators commute after time evolution [10, 11, 11–20]:

$$\mathcal{C}(r,t) \equiv \frac{1}{2}\text{Tr}\left(\rho\left[\mathcal{O}_0(t), \mathcal{O}_r\right]^\dagger \left[\mathcal{O}_0(t), \mathcal{O}_r\right]\right), \tag{1}$$

where $\mathcal{O}_r$ is an operator localized at position $r$. Expanding out the correlator gives a time ordered correlator (TOC) that is constant for Pauli-like observables and a nontrivial out-of-time-ordered correlator (OTOC). The quantity $\mathcal{C}(r,t)$ exhibits ballistic spreading and

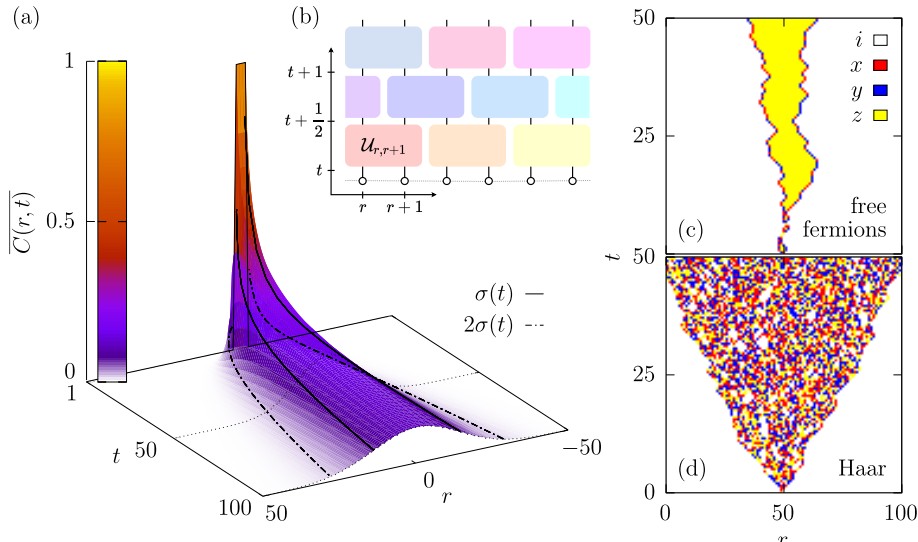

Figure 1: (a) Exact $\overline{\mathcal{C}(r,t)}$ measuring operator spreading in the $r - t$ plane for a system with 100 sites. The black curves envelop the $\sigma(t) = \sqrt{2t}$ and $2\sigma(t)$ regions. (b) Circuit scheme in the $r - t$ plane: random two-site unitaries $\mathcal{U}_{r,r+1}$ are applied to neighboring sites in a brick wall pattern. (c,d) Operator spreading can be understood in terms of the stochastic update of a single string of Pauli operators, $\mathcal{S}$. Starting with the string $\mathcal{S} = \ldots \otimes I \otimes Z \otimes I \otimes \ldots$, we show possible histories for its evolution under free fermion (c) and Haar-random unitary circuits (d) [1,3].

KPZ growth at the light cone interface [1,3], and the dynamics can be mapped to a biased random walk. In addition, one can map the evolution of the OTOC to that of a spreading ensemble of Pauli strings that rapidly scramble. In the presence of a conserved charge, the picture is modified owing to the diffusion of the conserved charges − there is a ballistic front that itself spreads diffusively [4,5]. The OTOC in such random unitary circuits provides a tractable instance of the universal physics of thermalization and the scrambling of quantum information in general non-integrable many-body interacting systems. The OTOC has also been used numerically to characterize information spreading in Hamiltonian evolution, both for interacting systems [21–25, 25] and for free fermions [26–29].

In this work we consider free fermion dynamics within the context of random unitary circuits. Free fermion circuits first appeared as classically simulatable matchgate circuits [30], which were later shown to correspond to a model of free fermions in 1D [31]. Unlike their interacting analogues, free fermion models with temporal noise exhibit diffusive dynamical features. This has been observed in the $\sim \sqrt{t}$ growth of entanglement entropy [2, 32–34] and in relaxation dynamics [32, 35, 36]. This diffusive dynamics sits in sharp contrast to the known free-fermion behaviors without temporal disorder − ballistic behavior for spatially non-disordered systems and Anderson localization (no spreading) for spatially ordered systems. Addition of temporal disorder renders either of these classes of systems diffusive.

Given the central role of free fermion systems in condensed matter theory, arriving at a general understanding of this remarkable diffusive behavior is clearly important. We present steps toward such a general understanding, by analyzing operator spreading and entanglement in the random circuit setting. Both spatially disordered and spatially uniform systems are readily treated in this framework. We adapt the Pauli string picture of operator spreading to free fermions, thus providing (i) a physical understanding of

how one obtains diffusion instead of ballistic behavior, and (ii) a precise picture of how free fermion circuits have only weak quantum information scrambling. In addition, an *exact* calculation of $\overline{\mathcal{C}(r, t)}$ is possible, where the overline indicates an average over random circuits. We thus provide an exact derivation of the previously observed diffusion and elucidate the slow scrambling of free fermion dynamics.

We consider three distinct instances of free fermion evolution: a non particle conserving spatio-temporal random circuit (NC-ST), its restriction to a particle conserving process (C-ST), and a spatially homogeneous case where randomness appears only in the time direction (NC-T). Each of these cases shows diffusive operator spreading in the OTOC, and correspondingly, entanglement entropy growing as a square root of time. We also show explicitly numerically that the phenomenon disappears without temporal disorder $-$ a non particle conserving circuit with quenched spatial disorder (NC-S) shows Anderson localization (see Section 5.12.1).

In the next section we describe the free fermion random circuit model that we explore later in the paper and the observables we consider. Then, in Section 3, we give a self-contained overview of all of our results before launching into a detailed presentation of the many-body and single-body calculations.

## 2  Free fermion circuit model and operator spreading

### 2.1  Fermionic quadratic Hamiltonians

Before describing the random circuit model in detail we briefly remind the reader of some facts about free (or quadratic) fermions.

Consider a free fermion (FF) chain with $L$ sites. The Hamiltonian has, at most, quadratic terms in the fermionic operators, $a$ and $a^\dagger$, and it can be written as

$$\mathcal{H} = \frac{1}{2} A^\dagger H A \qquad \text{with} \qquad H = \begin{pmatrix} h & \Delta \\ \Delta^\dagger & -h^T \end{pmatrix}, \tag{2}$$

where $A = (a_1, \cdots, a_L, a_1^\dagger, \cdots, a_L^\dagger)^T$ is the Nambu vector and $h = h^\dagger$ and $\Delta = -\Delta^T$. This system is invariant under the exchange of creation and annihilation operators, i.e. it is particle-hole symmetric, which translates to

$$SHS = -H^T \qquad \text{with} \qquad S = \begin{pmatrix} 0 & \mathbb{1}_{L \times L} \\ \mathbb{1}_{L \times L} & 0 \end{pmatrix}. \tag{3}$$

An operator $H$ respecting (3) is said to be particle-hole (PH) symmetric.

**Single-body description**   Given the $2L \times 2L$ single-body Hamiltonian matrix, $H$, we may diagonalize it and compute arbitrary observables in terms of the single-particle states. More specifically, having the two-point correlator $\chi = \langle AA^\dagger \rangle$, the mean value of any observable of the form $\mathcal{O} = A^\dagger O A$ can be computed using

$$\langle \mathcal{O}(t) \rangle = \text{tr}\left( \rho A^\dagger(t) O A(t) \right) = -\text{tr}\left( O \chi(t) \right), \tag{4}$$

with $\rho = \mathrm{e}^{-\beta \mathcal{H}}/Z$ and $Z = \text{tr}(\mathrm{e}^{-\beta \mathcal{H}})$ for a thermal density matrix (this is valid for the circuits considered with $\beta = 0$). Furthermore, we know that the fermionic operators $A(t)$ obey $\partial_t A(t) = -iHA(t)$, evolving according to $A(t) = \exp(-iHt)A_0$. With this, and

considering the initial $\chi = \langle AA^\dagger \rangle = \text{diag}\,(1 - n_1, \ldots, 1 - n_L, n_1, \ldots, n_L)$, we obtain the following equation for the evolution of $\chi(t)$:

$$\chi(t) = e^{-iHt}\chi e^{iHt}. \tag{5}$$

Hence, we have the tools to compute (4). In particular, the entanglement entropy itself can be computed in a simple manner with [37, 38]

$$S = -\text{tr}\,(\rho \log \rho) = -\text{tr}\,(\chi \log \chi). \tag{6}$$

## 2.2 Random circuit model

The random circuit acts on a chain of $L$ sites $(r = 0, \ldots, L - 1)$ with periodic boundary conditions as a discrete time protocol. A time step corresponds to one layer of random gates acting on even bonds, followed by one layer of random gates acting on odd bonds. A gate applied to neighboring sites acts non-trivially on the local 4-dimensional Hilbert space referring to these sites. The evolution follows the brick wall pattern of Fig. 1 (b).

### 2.2.1 Time evolution operator

Let us distinguish between the random unitary circuits used in the single and many-body basis. We use calligraphic and non-calligraphic letters to denote many (MB) and single-body (SB) evolution operators, respectively.

**Many-body basis** When working in the many-body basis, the $2^L \times 2^L$ circuit $\mathcal{U}$ evolving $|\Psi(t)\rangle = \mathcal{U}(t)|\Psi\rangle$ and $\mathcal{O}(t) = \mathcal{U}^\dagger(t)\mathcal{O}\mathcal{U}(t)$ is obtained from $q^2 \times q^2$ unitaries (with $q = 2$ the local Hilbert space dimension) as follows. A time step corresponds to the action of $\mathcal{U} = \mathcal{U}_{\text{even}}\mathcal{U}_{\text{odd}}$, where $\mathcal{U}_{\text{even}}$ and $\mathcal{U}_{\text{odd}}$ are built as

$$\mathcal{U}_{\text{even}} = \bigotimes_{R=0}^{L/2-1} \mathcal{U}_{2R,2R+1} \qquad \text{and} \qquad \mathcal{U}_{\text{odd}} = \bigotimes_{R=0}^{L/2-1} \mathcal{U}_{2R-1,2R}, \tag{7}$$

with $\mathcal{U}_{r,r+1}$ a two-site random unitary acting non-trivially on the local 4-dimensional Hilbert space of sites $r$ and $r + 1$.

**Single-body basis** In the single-body picture, each observable $O$ evolves according to $O(t) = U^\dagger(t)OU(t)$, with $U(t)$ the single-particle evolution operator. Each $U(t)$ is a succession of $t$ gates $U = U_{\text{even}}U_{\text{odd}}$, each composed of one even and one odd layers. Considering $U_{r,r+1}$ as the $2L \times 2L$ operator which acts non-trivially on sites $r$ and $r + 1$, even and odd layers can be written as

$$U_{\text{even}} = \prod_{R=0}^{L/2-1} U_{2R,2R+1} \qquad \text{and} \qquad U_{\text{odd}} = \prod_{R=0}^{L/2-1} U_{2R-1,2R}. \tag{8}$$

In its turn, $U_{r,r+1} = u_{r,r+1} + \bar{\mathbb{P}}_{r,r+1}$, where

$$u_{r,r+1} = \sum_{s,s'} \sum_{x,x'\in\{r,r+1\}} |x,s\rangle \langle x,s| u |x',s'\rangle \langle x',s'| \tag{9}$$

is the $4 \times 4$ matrix responsible for mixing the pair of sites $r$ and $r + 1$, while

$$\bar{\mathbb{P}}_{r,r+1} = \sum_{s} \sum_{x\notin\{r,r+1\}} |x,s\rangle \langle x,s| \tag{10}$$

is the trivial projector into the complement of the pair. Taking this into account, we can write each layer in terms of the non-trivial $u$s as

$$U_{\text{even}} = \sum_{R=0}^{L/2-1} u_{2R,2R+1} \qquad \text{and} \qquad U_{\text{odd}} = \sum_{R=0}^{L/2-1} u_{2R-1,2R}. \tag{11}$$

### 2.2.2 Free fermion gates

Depending on which gates are used to build the random circuit, it can mimic different types of dynamics. To simulate generic unitary dynamics, the gates considered are uniformly distributed, i.e. Haar-distributed, in the unitary group $\mathsf{U}(4)$. We say the circuits are composed of Haar-random unitaries [1,39]. To simulate free fermion dynamics, we must consider a more restricted set of gates in $\mathsf{U}(4)$ respecting the free fermion symmetries. Namely, the gates must preserve the particle-hole symmetry and conserve the parity of the number of particles.

We must realize both the many-body evolution operator, $\mathcal{U}(t) = \exp(-i\mathcal{H}t) = \exp(-i/2 A^\dagger H A t)$, and the single-body evolution operator, $u(t) = \exp(-iHt)$. These operators acting on $r$ and $r+1$ are related by

$$(A_r(t))_i = \mathcal{U}_{r,r+1}^\dagger (A_r)_i \mathcal{U}_{r,r+1} = (u_{r,r+1}(t)A_r)_i, \tag{12}$$

with $A_r = (a_r, a_{r+1}, a_r^\dagger, a_{r+1}^\dagger)^T$ the Nambu vector and $i = 1, \ldots, 4$ an index selecting a component of $A_r$.

We first realize $u$, from which we obtain $\mathcal{U}$ from $u$ using (12).

**Single-body basis**   The single-body evolution operator $u$ inherits the particle-hole symmetry (3) as $Su^TS = u^\dagger$. Going to the Majorana representation $u \to \tilde{u} = VuV^\dagger$, with $V$ defined in (13), one sees that particle-hole symmetry leads $\tilde{u}$ to be orthogonal, i.e $\tilde{u}\tilde{u}^T = \mathbb{1}$. This transformed back to the Dirac representation produces to the desired result:

$$u = V^\dagger O V \qquad \text{with} \qquad V = \frac{1}{\sqrt{2}} \begin{pmatrix} \mathbb{1}_{2\times2} & \mathbb{1}_{2\times2} \\ -i\mathbb{1}_{2\times2} & i\mathbb{1}_{2\times2} \end{pmatrix} \tag{13}$$

and $O \in \mathsf{O}(4)$ a random Haar-distributed orthogonal matrix. Such $O$ can be obtained by generating a $4 \times 4$ real matrix $z = qr$ belonging to the real Ginibre ensemble, and performing a unique QR decomposition [40]. This is done using Python's algorithm: `scipy.stats.ortho_group.rvs(4)` [41].

For the particle number conserving case we have $H = h \oplus -h^T$, i.e. there are no anomalous terms. In this case, $u_{r,r+1} = v_{r,r+1} \oplus v_{r,r+1}^*$ (where $\oplus$ stands for direct sum), with $v_{r,r+1} \in \mathsf{U}(2)$ a Haar distributed unitary matrix.

**Many-body basis**   Having $u$ given by (13), one obtains $\mathcal{U}$ implicitly via (12). These $\mathcal{U}$ gates are precisely the matchgates introduced in Ref. [30], whose connection with non-interacting fermions was later established [42].

### 2.3 Operator spreading

The time-ordered density-density correlator becomes trivial when averaged over temporal disorder (see Section 5.3), but $\mathcal{C}(r,t)$, introduced in (1), remains non-trivial upon averaging. We consider an average over separable initial states, which is equivalent to taking $\rho \propto 1$ in (1), i.e. the infinite temperature ensemble. In the following, we shall consider quadratic observables $\mathcal{O}_r = \Psi^\dagger O_r \Psi$ where $\Psi \equiv (a_1, \ldots, a_L, a_1^\dagger, \ldots, a_L^\dagger)^T$ and $O_r$ is a local single-particle operator.

# 3 Overview

This section is intended to spare readers the full details of the calculations by focussing on the key ideas, on the results and their implications. Given the background covered in Section 2, this section should be reasonably comprehensible without referring to later sections though, for more avid readers, we do indicate where further details are to be found.

## 3.1 Many-body calculation

We now sketch the central result of the paper − an exact many-body calculation of the OTOC. This calculation provides important intuition into the diffusive growth of entanglement and offers insight into how free fermion random circuits differ from their generic unitary counterparts. One of the outcomes of the calculation is a demonstration that free fermions scramble quantum information poorly in contrast to the unitary case.

The single-site Pauli operators are to be denoted $I, X, Y, Z$. We compute the OTOC for operator an $\mathcal{O}_0(t)$ in terms of a basis of Pauli strings $\mathcal{S}$: $\mathcal{O}_0(t) = \sum_\mathcal{S} a_\mathcal{S}(t)\mathcal{S}$ with normalization $\sum_\mathcal{S} a_\mathcal{S}^2(t) = 1$. The OTOC may then be written as $\mathcal{C}(r,t) = \sum_{\mathcal{S}:\mathcal{S}_r=X,Y} 2a_\mathcal{S}^2(t)$, where the nonvanishing contribution to the OTOC dictates that the Pauli string at position $r$ must be $S_r = X$ or $Y$. The problem of determining the OTOC at time $t$ is reduced to finding the dynamics of $\overline{a_\mathcal{S}(t)^2} = 4^{-L}\overline{\text{tr}\left(\mathcal{U}^\dagger \mathcal{O}_0(t-1/2)\mathcal{U}\mathcal{S}\right)^2}$, with $\mathcal{U} = \mathcal{U}_\text{even}$ or $\mathcal{U}_\text{odd}$. Expanding $\mathcal{O}_0(t-1/2)$ in a basis of Pauli strings yields

$$\overline{a_\mathcal{S}^2(t)} = \sum_{\mathcal{S}'\mathcal{S}''} \overline{a_{\mathcal{S}'}(t-1/2)a_{\mathcal{S}''}(t-1/2)W_{\mathcal{S}\leftarrow\mathcal{S}',\mathcal{S}''}}, \tag{14}$$

where $W_{\mathcal{S}\leftarrow\mathcal{S}',\mathcal{S}''} \equiv \prod_r \overline{\text{tr}(\mathcal{U}_{r,r+1}^\dagger S'_{r,r+1}\mathcal{U}_{r,r+1}S_{r,r+1})\text{tr}(\mathcal{U}_{r,r+1}^\dagger S''_{r,r+1}\mathcal{U}_{r,r+1}S_{r,r+1})}$, with $S_{r,r+1}$ the substring at sites $r$ and $r+1$.

Averaging over the many-body random unitary operators (see Section 4.1) results in

$$\overline{a_\mathcal{S}^2(t)} = \sum_{\mathcal{S}'} W_{\mathcal{S}\mathcal{S}'}\overline{a_{\mathcal{S}'}^2(t-1/2)}. \tag{15}$$

This result holds for both generic random unitary dynamics and the constrained free fermion dynamics considered here. The difference is reflected in the weights $W_{\mathcal{S}\mathcal{S}'}$, which depend on the distribution of $\mathcal{U}$ over the space of local unitaries.

For generic random unitaries, the weights for a two-site chain $\mathcal{S}$ are [1,3]

$$W_{\mathcal{S}\mathcal{S}'} = \delta_{\mathcal{S},I}\delta_{\mathcal{S}',I} + \frac{1}{15}\left(1 - \delta_{\mathcal{S},I}\right)\left(1 - \delta_{\mathcal{S}',I}\right), \tag{16}$$

meaning that a pair of identity operators on a pair of sites evolves into itself whereas the remaining 15 nontrivial Pauli string pairs are updated into any nontrivial pair with equal probability. From the initial condition with a single $Z$ operator on site 0, the operator spreads to the left and right. At the right-hand boundary separating a nontrivial string from unit operators, there is a bond with $S_r \otimes I$ with $S_r = X, Y$ or $Z$. The update rule implies that the $I$ will be turned into one of $X, Y$ or $Z$ more often than it remains the same − out of the 15 nontrivial substrings, 12 lead the right endpoint to propagate to the right. It follows that the evolution for generic unitaries is a biased random walk with a ballistically spreading boundary. Within the lightcone boundary, the operators update between all the available nontrivial configurations resulting in a scrambling of quantum information represented pictorially in Fig. 1(d). This figure shows the space-time evolution of a single realization of the Pauli string starting from a single $Z$ operator halfway along

the chain. As time passes, the operator spreads with a roughly linear front. The colors represent the different Pauli operators contributing to the string. From one time step to the next, the three nontrivial Pauli operators are uncorrelated on each site within the bulk of the string. In other words, they are scrambled completely.

The evolution of Pauli strings for the free fermion case is qualitatively different to the generic case. The update rules on a pair of sites preserve $I \otimes I$ and $Z \otimes Z$ strings. The remaining 14 operators fall into three classes that scramble among themselves with equal probability. These are

$$\{I \otimes Z, Z \otimes I, X \otimes X, X \otimes Y, Y \otimes X, Y \otimes Y\} \tag{17}$$

$$\{I \otimes X, I \otimes Y, X \otimes Z, Y \otimes Z\} \tag{18}$$

$$\{X \otimes I, Y \otimes I, Z \otimes X, Z \otimes Y\}. \tag{19}$$

Notice that each set contains either parity preserving or non-preserving substrings, such that parity is globally conserved, as required. Starting from the string with only $Z$ at site 0 nontrivial, it is straightforward to see that these update rules lead to strings with $X$ or $Y$ at separated boundaries with a frozen core of $Z$ operators. The chances are the same that the boundary moves outwards or inwards so each distinct boundary is a random walker that diffuses. The two ends can meet and restore the configuration with a single $Z$ operator. The intersection points are localized departures from the random walk and we explicitly compute these departures in the next section. A single realization of the Pauli string evolution just described is shown in Fig. 1(c). Unlike the corresponding figure for the generic unitary case, one sees that the boundaries are unbiased random walkers that are $X$ or $Y$ operators while, between the boundaries, the string is a block of $Z$ operators. We conclude that the free fermion random circuit evolution is diffusive. Besides, free fermions are ineffective at scrambling quantum information in the precise sense given by the string picture just described − the only interesting dynamics is that of the string endpoints, between which there is a frozen core of $Z$ operators.

## 3.2 Single-body calculation

In the previous subsection we presented an exact calculation of the OTOC in the many-particle basis that supplies a useful intuition about the nature of free fermions in relation to the spreading of quantum information. The computation of $\mathcal{C}(r,t)$ for free fermions − in common with other correlators [37, 38] − can be brought into a form where the trace need only be performed over $2L \times 2L$ matrices rather than over the entire Hilbert space. In Section 5, we provide an independent calculation of the OTOC from this single-particle perspective. One can show that the many-body correlator can be written in terms of single-body quantities as $\mathcal{C}(r,t) = 2[C_1(r,t) - C_2(r,t)]$, where

$$C_1(r,t) = \text{tr}\left(O_0^2(t)O_r^2\right), \tag{20}$$

$$C_2(r,t) = \text{tr}\left(O_0(t)O_rO_0(t)O_r\right), \tag{21}$$

and $O(t) = UO(t-1)U^\dagger$. The general relation between the single particle and many-body TOC and OTOC is given in Section 5.1. The result is that both $\overline{C_1}$ and $\overline{C_2}$ spread diffusively.

Exact results for $\overline{C(r,t)}$ are shown in Fig. 1(a) for a symmetrized particle number operator $\mathcal{O}_r = a_r^\dagger a_r - a_r a_r^\dagger = 2a_r^\dagger a_r - 1$ for a spatio-temporal random circuit that does not conserve particle number (NC-ST). Notice that Jordan-Wigner transforming $\mathcal{O}_r$ results in the Pauli operator $Z$ at site $r$ used in the many-body calculations. Whereas $\overline{C_1}$ in the continuum limit evolves as $\overline{C_1(x,t)} = (1/\sqrt{\pi t}) \exp(-x^2/4t)$, the contribution from $\overline{C_2}$ can

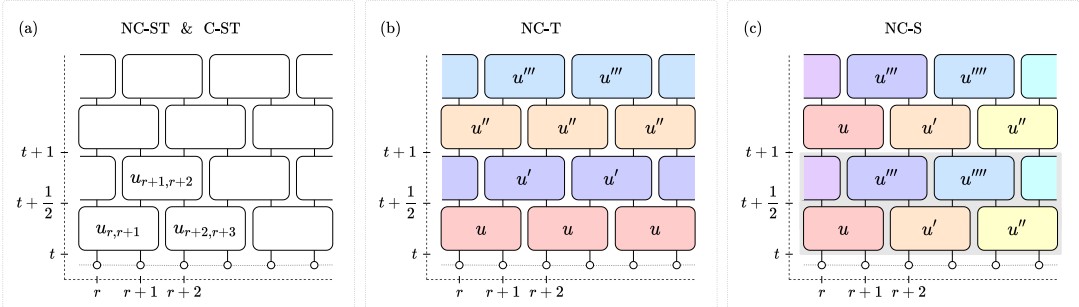

Figure 2: Scheme of a unitary circuit acting on $L$ sites: unitaries $u_{r,r+1}$ are applied to neighboring sites $r$ and $r+1$ in a brickwall pattern. At integer time steps, we apply even layers of gates, i.e. two-site unitaries are applied to even pairs of sites $(2R, 2R+1)$, with $R = 0, \ldots, L/2 - 1$; and, at half-integer time steps, we apply odd layers, with gates acting on odd pairs instead, i.e. on $(2R-1, 2R)$. Due to periodic boundary conditions, the total number of sites $L$ is even and the sites are acted upon by modulo $L$. (a) Circuit scheme for the NC-ST and C-ST cases in the $r - t$ plane, with randomness is space and time. The two cases differ in the gates used: the NC-ST case uses two-site free fermion gates $u_{r,r+1} = V^\dagger O V$, where $O$ is a $4 \times 4$ random Haar-distributed orthogonal matrix, while for C-ST $u_{r,r+1} = v_{r,r+1} \oplus v^*_{r,r+1}$, with $v_{r,r+1}$ a $2 \times 2$ random Haar-distributed unitary matrix. When working in the many-body basis instead, many-body gates $\mathcal{U}$ are applied. (b) Circuit scheme for the NC-T case. At each half time step, the same unitary $u$ is applied to all pairs of sites. (c) Circuit scheme for the NC-S case. One even and one odd layers are generated as usual, these are fixed and repeated in time, i.e. the grey box is repeated in time. In (b) and (c) the gates are identical to the ones used for NC-ST.

be viewed as arising from a two-dimensional diffusive process projected onto one dimension, $\overline{C_2(x,t)} \sim 1/(4\pi t) \exp(-x^2/2t)$ (see Section 5.5). These contributions have a natural interpretation in terms of the random walkers coming from the many-particle calculation. The term $\overline{C_1}$ is the part coming from the two independent random walkers while $\overline{C_2}$ is the sub-leading part originating from the interaction and annihilation of those walkers. Notice that $\lim_{t\to\infty} \overline{C_2}/\overline{C_1} = 0$, i.e. deviations from the random walk are subleading in time, as expected since the Pauli string grows in time such that its endpoints are less likely to meet.

### 3.3 Extensions to particle conserving (C-ST), space (NC-T) and time (NC-S) translation invariant cases

We have shown that both $\overline{C_1}$ and $\overline{C_2}$ spread diffusively for free fermions in 1D in the presence of spatio-temporal noise (NC-ST). We now consider exact calculations for two further cases: C-ST where the fermion particle number is conserved and each gate in the quantum circuit is chosen randomly, and NC-T where the unitary evolution is spatially homogeneous but where there is temporal noise − a single gate is chosen randomly at each time step and applied to all pairs of sites, as shown in Fig. 2(b). Section 5 lays out in detail exact calculations of $\overline{C_1}$ and $\overline{C_2}$ for the different instances of free fermion evolution just enumerated. The result is that there is diffusive spreading in all three cases with diffusion constants coinciding with those found for NC-ST. A comparison between exact and numerical results for $\overline{C_1}$ and $\overline{C_2}$ for NC-ST, C-ST and NC-T is shown in Fig. 3(a).

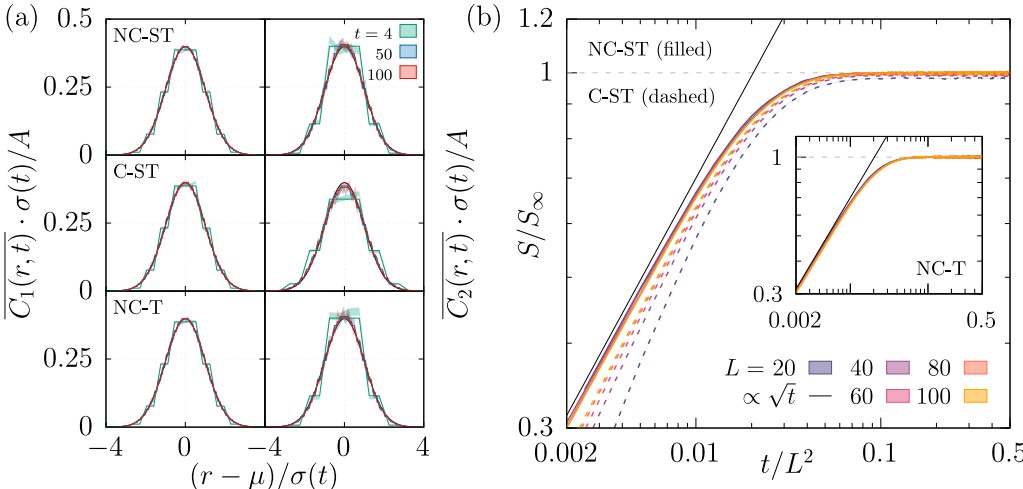

Figure 3: (a) Rescaled single-particle $\overline{C_1(r,t)}$ and $\overline{C_2(r,t)}$ for 100 sites subjected to different free-fermion random circuits, illustrating their diffusive evolution at different times. Both the analytical and numerical solutions, the latter including an average over 4000 disorder realizations, collapse to the continuum limit solution $g(r,t) = 1/(\sqrt{2\pi}) \exp[-(r-1/2)^2/(2\sigma^2(t))]$. (b) Growth of the entanglement entropy divided by the saturation value $S/S_\infty$ as a function of $t/L^2$ for random free fermions $-$ NC-ST (filled), C-ST (dashed) and NC-T (inset) $-$ with $L = 100$ and a subsystem of 50 sites showing $S(t) \propto \sqrt{t}$ at the earliest times. The data results from averaging over 1000 realizations up to $t = 5000$, starting from a random product state with particle number fixed to $L/2$. The curves for NC-ST and NC-T overlap.

In contrast, numerical results obtained for the temporal homogeneous case (NC-S) presented in Section 5.12.1, where even and odd layers of random gates are fixed and applied repeatedly in time - see Fig. 2(c) - show that $\overline{C_1}$ and $\overline{C_2}$ remain Anderson localized (see Fig. 13), decaying exponentially around $r = 0$ [43, 44].

### 3.4 Dynamics of entanglement

For completeness, we analyse the dynamics of the von Neumann entropy, $S = -\operatorname{tr}(\tilde{\rho} \ln \tilde{\rho})$, with $\tilde{\rho}$ the reduced density matrix of a subsystem of size $L/2$, starting from an initial product state with a well defined particle number. Fig. 3(b) shows that for the three processes the entanglement grows as $\sim D_S\sqrt{t}$ for small times (with $D_S$ a time independent constant), saturating at times $t_{\text{sat}} \sim L^2$. For $t \gg t_{\text{sat}}$, the saturation value $S_\infty = s_0 L + s_1 + O(1/L)$, coincides with the mean entanglement entropy of a random Gaussian state [45–49], with $s_0 \simeq 0.193$ for all the considered free fermion processes and $s_1 \simeq 0.085$ for the NC-ST case, well below the Page value ($s_0 = \ln 2/2 \simeq 0.346, s_1 = -1/2$) obtained by averaging over the full Hilbert space [9]. These results show that the rate of increase of the entanglement is compatible with the diffusion of quantum information we derived for the OTOC.

## 4 Operator spreading in the many-body basis

We now turn to the detailed calculations of the OTOC in the many-body (Section 4) and single-body basis (Section 5).

In this section, we present detailed calculations showing operator diffusion for free

fermion circuits. These calculations in the many-body basis provide an intuitive picture for diffusion in terms of a spreading Pauli string whose endpoints are approximate random walkers.

These calculations draw inspiration from recent works where Nahum *et al.* [1] and Keyserlingk *et al.* [39] obtained an analytical expression showing ballistic operator growth in Haar-distributed random circuits. We begin by summarizing these calculations. Then, we highlight the differences occurring for free fermion evolution.

## 4.1 Operator growth as spreading Pauli strings

The spreading of operators can be measured by the degree of non-commutativity between two local observables $\mathcal{O}_0$ and $\mathcal{O}_r$ centred, respectively, around positions $0$ and $r$ with finite support, i.e. by (1), which can be rewritten as

$$\mathcal{C}(r,t) = \frac{1}{2}\langle |[\mathcal{O}_0(t), \mathcal{O}_r]|^2\rangle = -\frac{1}{2}\langle [\mathcal{O}_0(t), \mathcal{O}_r]^2\rangle, \tag{22}$$

where the last equality holds for hermitian operators, i.e. $\mathcal{O}_r^\dagger(t) = \mathcal{O}_r(t)$.

We consider $\mathcal{O}_r \equiv \mathcal{O}_r(t=0) = \mathcal{Z}_r$ and $\mathcal{O}_0 \equiv \mathcal{O}_0(t=0) = \mathcal{Z}_0$, with

$$\mathcal{Z}_r = \cdots \otimes I_{r-1} \otimes Z_r \otimes I_{r+1} \otimes \cdots \in \mathsf{SU}(q^L) \tag{23}$$

acting with the single-site Pauli operator $Z \in \mathsf{SU}(q)$ on position $r$ and with the identity $I$ elsewhere. Here, $q = 2$ is the local Hilbert space dimension of a fermion. We will use the following notation for Pauli operators interchangeably: $\{I, X, Y, Z\} \equiv \{\sigma_0, \sigma_1, \sigma_2, \sigma_3\}$.

Let us decompose the evolving operator $\mathcal{Z}_0(t)$ in terms of strings $\mathcal{S}$ of $L$ single-site Pauli operators, i.e. $\mathcal{S} \in P_L = \{\sigma_{i_1} \otimes \ldots \otimes \sigma_{i_L} : i_1, \ldots, i_L = 0, 1, 2, 3\}$, which is a basis of $\mathsf{SU}(q^L)$,

$$\mathcal{Z}_0(t) = \mathcal{U}(t)^\dagger \mathcal{Z}_0 \mathcal{U}(t) = \sum_{\mathcal{S} \in P_L} a_{\mathcal{S}}(t)\mathcal{S}. \tag{24}$$

The normalization

$$\langle \mathcal{S}\mathcal{S}'\rangle = \mathrm{tr}(\rho_\infty \mathcal{S}\mathcal{S}') = \delta_{\mathcal{S}\mathcal{S}'} \tag{25}$$

follows from

$$\sigma_\mu \sigma_\nu = \mathbb{1}\delta_{\mu\nu} + i\varepsilon_{\mu\nu\alpha}\sigma_\alpha, \tag{26}$$

with $\nu, \mu, \alpha = 1, 2, 3$ and $\varepsilon_{\mu\nu\alpha}$ the Levi-Civita symbol.

Considering (22) to (26), we obtain [1, 39]

$$\mathcal{C}(r,t) = -\frac{1}{2}\langle [\mathcal{Z}_0(t), \mathcal{Z}_r]^2\rangle = \sum_{\mathcal{S} \in P_L : S_r = x, y} 2a_{\mathcal{S}}(t)^2, \tag{27}$$

where we considered the reduced density matrix to be in the infinite temperature Gibbs state $\rho = \rho_\infty = 1/2^L$, since we perform an average over separable initial states. Notice that the strings $\mathcal{S}$ contributing to $\mathcal{C}(r,t)$ must have either $X$ or $Y$ at position $r$ such that they do not commute with $\mathcal{Z}_r$, i.e. they belong to the set

$$\{\sigma_{i_1} \otimes \ldots \otimes \sigma_{i_L} : i_1, \ldots, i_{r-1}, i_{r+1}, \ldots, i_L = 0, 1, 2, 3 \wedge i_r = 1, 2\}. \tag{28}$$

Thus, determining the evolution of $\mathcal{C}(r,t)$ reduces to determining that of the coefficients $a_S(t)^2$.

**Evolution of the operator string distribution**    Let us obtain a recursive evolution equation for $a_S(t)^2$.

Using $a_S(t) = \langle \mathcal{Z}_0(t)S \rangle = q^{-L} \operatorname{tr}(\mathcal{Z}_0(t)S)$ and (24), it follows that

$$
\begin{aligned}
a_S(t)^2 &= \frac{1}{q^{2L}} \operatorname{tr}(\mathcal{U}^\dagger \mathcal{Z}_0(t-1/2)\mathcal{U}S)^2 \\
&= \frac{1}{q^{2L}} \sum_{\mathcal{S}',\mathcal{S}'' \in P_L} a_{\mathcal{S}'}(t-1/2) a_{\mathcal{S}''}(t-1/2) \operatorname{tr}(\mathcal{U}^\dagger S' \mathcal{U}S) \operatorname{tr}(\mathcal{U}^\dagger \mathcal{S}'' \mathcal{U}S) \\
&= \frac{1}{q^{2L}} \sum_{\mathcal{S}',\mathcal{S}'' \in P_L} a_{\mathcal{S}'}(t-1/2) a_{\mathcal{S}''}(t-1/2) \\
&\quad \times \prod_r \operatorname{tr}(\mathcal{U}^\dagger_{r,r+1} S'_{r,r+1} \mathcal{U}_{r,r+1} S_{r,r+1}) \operatorname{tr}(\mathcal{U}^\dagger_{r,r+1} S''_{r,r+1} \mathcal{U}_{r,r+1} S_{r,r+1}),
\end{aligned}
\tag{29}
$$

where $\mathcal{U}(t,t-1/2) = \mathcal{U}$ is either $\mathcal{U}_{\text{even}}$ or $\mathcal{U}_{\text{odd}}$ given by (7), with $\mathcal{U}_{r,r+1} \in \mathsf{U}(4)$ acting on sites r and r+1, and $\mathcal{S}_r = \bigotimes_r S_r = \cdots \otimes S_{r-2,r-1} \otimes S_{r,r+1} \otimes S_{r+2,r+3} \otimes \cdots \in \mathsf{SU}(2^L)$, with $S_{r,r+1} \in \mathsf{SU}(4)$ acting on sites $r$ and $r+1$. Moreover, notice we have used

$$
(\bigotimes_r \mathcal{U}_r)(\bigotimes_{r'} \mathcal{S}_{r'}) = \bigotimes_r \mathcal{U}_r \mathcal{S}_r
\tag{30}
$$

and $\operatorname{tr}(\bigotimes_r U_r S_r) = \prod_r \operatorname{tr}(U_r S_r)$.

In fact, we are interested in $a_S(t)^2$ averaged over circuit disorder, i.e.

$$
\begin{aligned}
\overline{a_S(t)^2} &= \frac{1}{q^{2L}} \sum_{\mathcal{S}'\mathcal{S}''} \overline{a_{\mathcal{S}'}(t-1/2) a_{\mathcal{S}''}(t-1/2)} \\
&\quad \times \prod_r \overline{\operatorname{tr}(\mathcal{U}^\dagger_{r,r+1} S'_{r,r+1} \mathcal{U}_{r,r+1} S_{r,r+1}) \operatorname{tr}(\mathcal{U}^\dagger_{r,r+1} S''_{r,r+1} \mathcal{U}_{r,r+1} S_{r,r+1})},
\end{aligned}
\tag{31}
$$

where averages at different time steps and sites can be performed independently since different unitaries are completely uncorrelated.

To compute the average over unitaries in (31) reduces to calculating the second moment of a many-body free fermion unitary $\mathcal{U}$, i.e. $\mathbb{E}_\mu(\mathcal{U} \otimes \mathcal{U} \otimes \mathcal{U}^* \otimes \mathcal{U}^*) \equiv \overline{\mathcal{U} \otimes \mathcal{U} \otimes \mathcal{U}^* \otimes \mathcal{U}^*}$, with $\mu$ the uniform measure over the group of unitaries $\mathcal{U}$ composing the circuit. The average over time evolution operators $\mathcal{U}$ uniformly distributed over the group of many-body free fermion operators, we denote by $\mathsf{U}_{\text{FFMB}}(4)$, and not $\mathsf{U}(4)$ is precisely where the calculation for the many-body free fermion case diverges from that for Haar unitaries [1, 39].

## 4.2   Operator spreading for free fermions

While the previous calculations are valid for any local circuit, henceforth the calculations are specific to free fermion circuits, with $\mathcal{U} \in \mathsf{U}_{\text{FFMB}}(4)$. We start by computing the second moment of a free fermion many-body unitary, and then resume computing (31) and the averaged OTOC, $\overline{\mathcal{C}(r,t)}$.

### 4.2.1   Second moment of many-body free fermion unitaries

We have established that, for $\mathcal{U}$ uniformly distributed in $\mathsf{U}_{\text{FFMB}}(4)$, the following expression holds:

$$
\begin{aligned}
\overline{\mathcal{U}_{a'a} \mathcal{U}^*_{b'b} \mathcal{U}_{c'c} \mathcal{U}^*_{d'd}} =&\ \left(\frac{1}{4} - \frac{1}{12}(\delta_{c\bar{a}} + \delta_{ac})\right) \delta_{ab}\delta_{cd}\delta_{a'b'}\delta_{c'd'}(\delta_{\bar{a}a'} + \delta_{aa'})(\delta_{\bar{c}c'} + \delta_{cc'}) \\
&+ \left(\frac{1}{4} - \frac{1}{12}(\delta_{b\bar{a}} + \delta_{ab})\right) \left(\delta_{c\bar{a}}\delta_{d\bar{b}}\delta_{\overline{a'}c'}\delta_{\overline{b'}c'}(\delta_{aa'} - \delta_{\bar{a}a'})(\delta_{bb'} - \delta_{\bar{b}b'})\right. \\
&\left.+ \delta_{ad}\delta_{bc}\delta_{a'c'}\delta_{b'c'}(\delta_{\bar{a}a'} + \delta_{aa'})(\delta_{\bar{b}b'} + \delta_{bb'})\right),
\end{aligned}
\tag{32}
$$

where $a, a', \ldots = 1, 2, 3, 4$ index the entries $\mathcal{U}_{a'a} = \langle a'|\mathcal{U}|a\rangle$ and where the bar operator indicates the conjugate entry in the same parity sector, i.e. $\bar{1} = 4$, $\bar{4} = 1$, $\bar{2} = 3$ and $\bar{3} = 2$, with entries 1 and 4 belonging to the even sector and 2 and 3 to the odd sector.

Expression (32) was obtained from numerical evidence as follows:

1. the relation between the many-body $\mathcal{U}$ and the single-body $u$ evolution operators (12) and unitarity, $\mathcal{U}^\dagger \mathcal{U} = \mathbb{1}$, allow us to obtain a non-linear expression for $\mathcal{U}$ in terms of the entries of $u$;

2. after generating $u$ as indicated in (13), we obtain $\mathcal{U}$ and $\overline{\mathcal{U} \otimes \mathcal{U} \otimes \mathcal{U}^* \otimes \mathcal{U}^*}$ numerically;

3. averaging over several realizations of $\mathcal{U}$, we identify the non-zero entries and their value, obtaining (32).

### 4.2.2   Operator string probability distribution

Consider the partition $\mathcal{A} = \{A_1, A_Z, A_I, A_L, A_R\}$ of the 16 two-site Pauli gates, $P_2$, with

$$A_1 = \{I \otimes I\}, \quad A_Z = \{Z \otimes Z\} \to \text{ trivial} \tag{33}$$

$$A_I = \{I \otimes Z, Z \otimes I, X \otimes X, X \otimes Y, Y \otimes X, Y \otimes Y\} \to \text{ walkers meet} \tag{34}$$

$$A_L = \{I \otimes X, I \otimes Y, X \otimes Z, Y \otimes Z\} \to \text{ left boundary } \sim \text{ random walker} \tag{35}$$

$$A_R = \{X \otimes I, Y \otimes I, Z \otimes X, Z \otimes Y\} \to \text{ right boundary } \sim \text{ random walker} \tag{36}$$

For $\mathcal{U}$ a MB free fermion gate, $\overline{(\mathcal{U}_{r,r+1})_{a'a}(\mathcal{U}^*_{r,r+1})_{b'b}(\mathcal{U}_{r,r+1})_{c'c}(\mathcal{U}^*_{r,r+1})_{d'd}}$ is given by (32), such that

$$\overline{\text{tr}(\mathcal{U}^\dagger_{r,r+1}S'_{r,r+1}\mathcal{U}_{r,r+1}S_{r,r+1})\,\text{tr}(\mathcal{U}^\dagger_{r,r+1}S''_{r,r+1}\mathcal{U}_{r,r+1}S_{r,r+1})} = 16\,\delta_{S'_{r,r+1}S''_{r,r+1}}W_{S_{r,r+1}S'_{r,r+1}} \tag{37}$$

with

$$W_{S_{r,r+1}S'_{r,r+1}} = \sum_{A\in\mathcal{A}}\sum_{s,s'\in A}\frac{\delta_{S_{r,r+1}s}\delta_{S'_{r,r+1}s'}}{\dim(A)} \tag{38}$$

and $\dim(A) = \sum_{s\in A} 1$ the number of elements in the set $A$.

Considering (37), (31) becomes

$$\overline{a_{\mathcal{S}}(t)^2} = \sum_{\mathcal{S}'} W_{\mathcal{S}\mathcal{S}'}\overline{a_{\mathcal{S}'}(t-1/2)^2} \quad \text{with} \quad W_{\mathcal{S}\mathcal{S}'} = \prod_r W_{S_{r,r+1}S'_{r,r+1}}. \tag{39}$$

i.e. a linear evolution equation for $\overline{a_{\mathcal{S}}(t)^2}$.

The circuit's structure is such that neighboring pairs of sites interact and different pairs are updated independently. The probability for $S'_{r,r+1}$ to be update to $S_{r,r+1}$ is $W_{S_{r,r+1}S'_{r,r+1}}$, with $\sum_{S_{r,r+1}\in P_2} W_{S_{r,r+1},S'_{r,r+1}} = 1$. According to (38) and the partition of $P_2$ given by (33) to (36), $W_{S_{r,r+1}S'_{r,r+1}}$ is such that an element $S_{r,r+1} \in A_k$ is mapped with uniform probability onto an element of $A_k$, with $k = 1, Z, I, L, R$. Then, the probability for $\mathcal{S}$ to be updated to $\mathcal{S}'$ is simply $W_{\mathcal{S}\mathcal{S}'} = \prod_r W_{S_{r,r+1},S'_{r,r+1}}$, with $\sum_{\mathcal{S}\in P_L} W_{\mathcal{S}\mathcal{S}'} = 1$. Hence, we must simply update the operators at each pair of interacting sites according to the probabilities $W_{S_{r,r+1},S'_{r,r+1}}$ given by (38), alternating between updating even and odd pairs, i.e. following the circuit's brick wall pattern.

While $\mathcal{Z}_0(t)$ is a superposition of exponentially many strings, with $a_{\mathcal{S}}(t)$ the weight with which string $\mathcal{S}$ appears in the superposition, the averaged $\mathcal{C}(r,t)$, which 'inherits' the

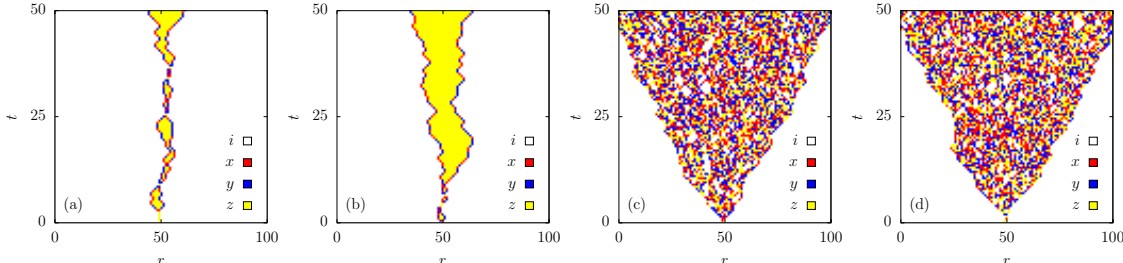

Figure 4: (a,b) Evolution of the initial operator string $\mathcal{S} = \ldots \otimes \mathbb{1} \otimes (\sigma_3)_{r=50} \otimes \mathbb{1} \otimes \ldots$ under free fermion dynamics. In one step, the operator string $\mathcal{S}$ is updated to $S'$ with probability $W_{\mathcal{S}\mathcal{S}'}$ given by (39). (c,d) Equivalent picture for string evolution under Haar-random unitary circuit. In this case, the probability for a two-site substring $\mathcal{S}$ to become $\mathcal{S}'$ is $W_{\mathcal{S}\mathcal{S}'}$ given by (16) [1, 39].

dynamics of $\mathcal{Z}_0(t)$, can be described under (27) and (39) as a stochastic update of a single string onto a single string. Consider the probability space $(\mathcal{S}, \overline{a_{\mathcal{S}}(t)^2})$, with $\overline{a_{\mathcal{S}}(t)^2}$ obeying $\sum_{\mathcal{S}} \overline{a_{\mathcal{S}}(t)^2} = 1$ the probability for the string $\mathcal{S}$ to appear, which evolves in time with (38). This probability space restricted to $\mathcal{S}$ with $S_r = X, Y$ describes the strings contributing to $\mathcal{C}(r, t)$.

### 4.2.3  Description of string evolution

We saw that the two-site string $S_{r,r+1}$ on sites $r$ and $r+1$ is updated to $S'_{r,r+1} \in P_2$ with probability $W_{S_{r,r+1}S'_{r,r+1}}$ given by (38). Namely, $W_{S_{r,r+1}S'_{r,r+1}}$ is such that an element $S_{r,r+1} \in A_k$ is mapped with uniform probability onto an element of $A_k$, with $k = 1, Z, I, L, R$ and $A_i$ given by (33) to (36). In particular, $I \otimes I$ and $Z \otimes Z$ each evolve trivially onto themselves. Although a two-site substring is mapped to a two-site substring in the same subset $A_i$, the circuit's brick wall pattern leads all the different subsets $A_i$ to be visited.

Let us anticipate that, starting with $\ldots \otimes I \otimes \sigma_3 \otimes I \otimes \ldots$, these rules allow the following operator string configurations

$$\ldots \otimes I \otimes \sigma_3 \otimes I \otimes \ldots \tag{40}$$

$$\ldots \otimes I \otimes \sigma_i \otimes \sigma_j \otimes I \otimes \ldots \tag{41}$$

$$\ldots \otimes I \otimes \sigma_i \otimes \sigma_3 \otimes \ldots \otimes \sigma_3 \otimes \sigma_j \otimes I \otimes \ldots \tag{42}$$

where $i, j = 1, 2$. Fig. 4(a,b) shows two possible string evolution histories or configurations obbeying these rules.

Consider we start with $\ldots \otimes I \otimes \sigma_3 \otimes I \otimes \ldots$, i.e. $\mathcal{Z}_0$. Only the pair involving $\sigma_3$ evolves non-trivially:

$$\underbrace{(I \otimes \sigma_3)} \vee \underbrace{(\sigma_3 \otimes I)} \rightarrow (\sigma_3 \otimes I) \vee (I \otimes \sigma_3) \vee (\bigvee_{i,j=1,2} \sigma_i \otimes \sigma_j), \quad \text{each with probability } 1/6,$$
$$\tag{43}$$

where the underbrackets link interacting sites which are updated to one of the right hand-side pairs with the given probability. This is, $\sigma_3$ can remain at the same location or move to the left or right, or $\sigma_i \otimes \sigma_j$ $(i, j = 1, 2)$ can arise from it. Having updated the even (odd) pairs, we must now update the odd (even) pairs. We just saw how $I \otimes \sigma_3$ and $\sigma_3 \otimes I$

evolve, $\sigma_i \otimes \sigma_j$ evolves as follows:

$$\bigvee_{i,j=1,2} I \otimes \sigma_i \otimes \sigma_j \otimes I \to \bigvee_{k,l=1,2} (I \otimes \sigma_k \vee \sigma_k \otimes \sigma_3) \otimes (\sigma_l \otimes I \vee \sigma_3 \otimes \sigma_l), \qquad (44)$$

each with probability 1/16 i.e. the string can remain of type (41) or it can 'grow' to the shape (42) which at later times will prevail. In the next time step, a possible update is

$$\bigvee_{i,j=1,2} \sigma_i \otimes \sigma_j \to (\sigma_3 \otimes I) \vee (I \otimes \sigma_3) \vee (\bigvee_{k,l=1,2} \sigma_k \otimes \sigma_l), \quad \text{each with probability } 1/4 \quad (45)$$

i.e. (41) can shrink back to (40), i.e. a single $\sigma_3$ operator, or keep expanding to (42). Analogously, (42) can shrink back to (41) or continue expanding and keep its shape (42). The tendency will be for the string to grow.

Summarizing, in general we have a string of single-qubit $\sigma_3$ operators with either a $\sigma_1$ or a $\sigma_2$ operator at each of its endpoints. Hence, the interesting behaviour occurs at the string endpoints.

Let us focus on the behaviour of the endpoints, we define as the nontrivial operators (i.e. $\neq I$) furthest to the left and to right of the origin. While the subsets $A_1$ and $A_Z$ (33) of $P_2$ correspond to trivially updating the string outside and inside its non-trivial region, $A_L$ (35) and $A_R$ (36) correspond to, respectively, the left and right endpoints moving independently when far apart. Note that, although the two-site gates in $A_L$ and $A_R$ do not preserve the parity of particle number locally, this is preserved globally. Moreover, these sets include the (locally) non-conserving gates such that a non-conserving gate is mapped to a non-conserving gate, as required to globally preserve the particle number parity. Finally, $A_I$ (34) corresponds to the two endpoints meeting and occasionally annihilating.

Consider that the two endpoints are far apart, neglecting their interaction via the substrings in $A_I$ (34) that can occur. The substrings including, for example, the right endpoint are those in $A_R$ (36), which are equally probable to appear. Averaging over these endpoint configurations, it happens that the averaged endpoint describes a random walk: it has a 1/4 probability of moving one unit to the left or right in a half-time step and a 1/2 probability of remaining in the same place. This initial average compensates for the fact that half of the walkers are biased to the left and the other half to the right, because of the circuit's brick wall pattern: updating $Z \otimes X$ or $Z \otimes Y$ results in the walker remaining in the same place or moving to the left, while updating $X \otimes 1$ or $Y \otimes 1$ results in the walker remaining in the same place or moving to the right. Thus,

1. when far apart, the left and right endpoints $\sigma_1$ or $\sigma_2$ are $\sim$ 1D independent random walkers,

2. when nearby, the walkers interact (e.g. they annihilate). This results in sub-leading corrections $\sim O\left(1/\sqrt{t}\right)$ to the random walk.

### 4.2.4 String endpoints as approximate random walkers

Remember the connection between $\overline{\mathcal{C}(r,t)}$ and the picture of a spreading string given by (24): the correlator $\overline{\mathcal{C}(r,t)}$ gets contributions $2\overline{a_{\mathcal{S}}(t)^2}$ from strings $\mathcal{S}$ with endpoints $\sigma_1$ or $\sigma_2$ at site $r$. Hence, the endpoint behaviour is truly what defines the free fermion OTOC.

Since we know the rules to obtain all possible string configurations, such as the ones shown in Fig. 4(a,b), to sum over all of these yields $\overline{\mathcal{C}(r,t)}$. In Fig. 5(a), we show $\overline{\mathcal{C}(r,t)}$ for free fermion circuits obtained by considering $10^5$ random realizations of string evolution histories, such as seen in Fig. 4(a,b). Since each of the string endpoints approximately

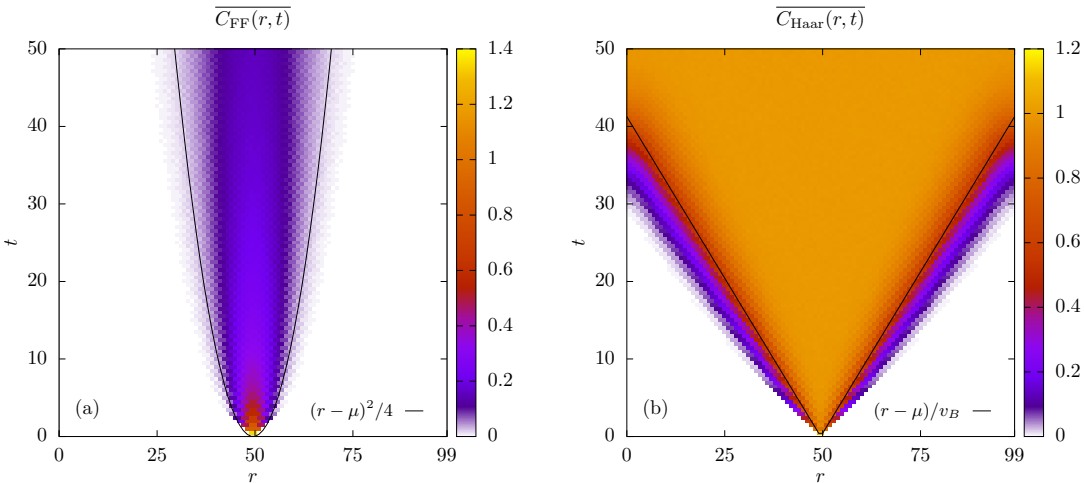

Figure 5: $\overline{\mathcal{C}(r,t)}$ obtained for (a) free fermion evolution and (b) evolution under a circuit of Haar-random unitaries [1,39] for a system with $L = 100$ and periodic boundary conditions. This was obtained numerically by considering the contribution of $10^5$ randomly generate string evolution histories, such as seen in Fig. 4. While for the chaotic case the OTOC spreads ballistically, with $v_B = 6/5$ the front speed set by the circuit geometry [1,39], under free fermion evolution it diffuses with $\sigma \sim \sqrt{2t}$.

describes a 1D random walk, as discussed at the end of last section, $\overline{\mathcal{C}(r,t)}$ is approximately given by twice the probability for a random walker to be at position $r$ after time $t$. On its turn, the probability for a random walker starting at $r = 0$ to be at position $r$ after time $t$ is

$$P(r,t) = \frac{1}{2^t}\binom{n}{(r+t)/2} \sim \overline{\mathcal{C}(r,t)}/2. \tag{46}$$

## 5 Operator spreading in the single-body basis

The exact many-body calculation of the free fermion OTOC was carried out in the previous section. Here, the same exact result is obtained using instead the single-body basis. While the many-body result has an intuitive interpretation in terms of spreading strings with approximate random walkers at the endpoints, it is convenient to compute the deviations to exact random walkers occurring when the string endpoints meet using the single-body picture. Such calculations run over many pages so, here, we give a summary of the main steps to guide the reader through the remainder of the section.

The analytical calculation of the OTOC given here makes use of the single-body formalism introduced in Section 2.1. We study the correlator of commutators $\mathcal{C}(r,t)$, introduced in (1), that is written in terms of a time-ordered correlation function (TOC), $\mathcal{C}_1(r,t)$, and an out-of-time order correlator, $\mathcal{C}_2(r,t)$. In this section, we analyse these in the local single particle basis. It turns out (Section 5.1) that the many-body correlator (1) can be written in terms of single particle correlators $C_1(r,t)$ and $C_2(r,t)$, which are similar respectively to $\mathcal{C}_1(r,t)$ and $\mathcal{C}_2(r,t)$. However, despite the similarity, the correspondence between $C_1(C_2)$ and $\mathcal{C}_1(\mathcal{C}_2)$ is not one-to-one, with the single particle TOC and OTOC being a mixture of the many-body TOC and OTOC. In this section, when referring to TOC and OTOC we usually mean the single-body ones.

Observables for the random circuit must be disorder averaged. A summary of results on random matrix averages, needed for the average of free fermion gates, is given in Section 5.2 and further explained throughout the text. One can imagine building the random circuit for a given disorder realization, computing the desired correlator and then averaging over realizations. This computation can be done instead by studying first the action of a single layer, acting on even or odd pairs of sites, and then building up to multiple layers. Before proceeding to calculate higher order correlators, we briefly show that the time-ordered density-density correlator becomes trivial when averaged over temporal disorder (Section 5.3). Section 5.4 presents the calculation of the time-ordered correlator $C_1$ for the NC-ST case, where we begin by establishing a vectorized notation to be used throughout the text. The calculation of $\overline{C_1}$ involves an average over two unitaries (Section 5.4) and shows the TOC to be given by a discrete random walk in 1D (Section 5.4). Taking the continuum limit of the discrete evolution equation for $\overline{C_1}$ results in a diffusion equation in 1D (Section 5.4) and a Gaussian broadening with standard deviation $\sigma(t) = \sqrt{2t}$. In Section 5.5, the calculation of $\overline{C_2}$ proceeds analogously to that of $\overline{C_1}$ but is more involved not least because the disorder average is carried out over a product of four unitaries rather than two in the case of $\overline{C_1}$. The result is that $\overline{C_2}$ is the diagonal of a two-dimensional quantity whose discrete evolution equation, detailed in Section 5.5, is approximated by a 2D diffusion equation in the continuum limit, and the OTOC is approximated by a Gaussian with broadening width $\sigma(t) = \sqrt{t}$ (Section 5.5). After establishing the results for the NC-ST case, we extend these to the NC-T and C-ST cases by pointing out the differences with regards to NC-ST. The different structure of the NC-T circuit plays no role and both $\overline{C_1}$ (Section 5.6) and $\overline{C_2}$ (Section 5.7) are exactly equal to the ones obtained for NC-ST. The same holds true for $\overline{C_1}$ in the C-ST case (Section 5.8). However, the conserving character of the unitaries used to build the circuit leads $\overline{C_2}$ to be different from its non-conserving counterpart. The details given in Section 5.9 do not alter the qualitative behaviour in the continuum limit, affecting only the normalization of the Gaussian obtained before.

Lastly, the analytical results obtained for the three instances of free fermion evolution (NC-ST, NC-T and C-ST) are shown to agree with simulations and also with the continuum limit solutions (Section 5.12). Furthermore, before concluding we present numerical results for a non particle conserving circuit with quenched spatial disorder (NC-S) that Anderson localizes (Section 5.12.1). Finally, we compare the MB and SB results and revisit how the OTOC deviates from a 1D random walk (Section 6).

## 5.1 TOC and OTOC in the single-body picture

The spreading of operators can be measured by (1), we rewrite here considering two local observables $\mathcal{W}$ and $\mathcal{V}$ centred, respectively, around positions 0 and $r$ with finite support:

$$\mathcal{C}(r,t) = \frac{1}{2}\langle |[\mathcal{W}(t), \mathcal{V}]|^2 \rangle = \mathcal{C}_1(r,t) - \mathcal{C}_2(r,t), \tag{47}$$

where $\mathcal{W} = \mathcal{U}^\dagger(t)\mathcal{W}\mathcal{U}(t)$ and the TOC and OTOC are, respectively, $\mathcal{C}_1(r,t) = \langle \mathcal{W}^2(t)\mathcal{V}^2 \rangle$ and $\mathcal{C}_2(r,t) = \langle \mathcal{W}(t)\mathcal{V}\mathcal{W}(t)\mathcal{V} \rangle$. Quadratic many-body observables $\mathcal{V} = \mathcal{V}(0)$ and $\mathcal{W} = \mathcal{W}(0)$ can be written as $\mathcal{V} = A^\dagger V A$ and $\mathcal{W} = A^\dagger W A$, with $A = (a_1, \cdots, a_L, a_1^\dagger, \cdots, a_L^\dagger)^T$ the Nambu vector and $a_r$ and $a_r^\dagger$ the fermionic operators at position $r$. Considering that $\mathcal{W}$ and $\mathcal{V}$ have this form, we can express (47) in terms of $V$ and $W$, the so called single-body observables. For this, it is convenient to write the relation between many and single-body operators

$$\mathcal{V} = \partial_v\big|_{v=0} \exp\left(vA^\dagger V A\right) \qquad \text{and} \qquad \mathcal{W} = \partial_w\big|_{w=0} \exp\left(wA^\dagger W A\right) \tag{48}$$

This operator evolves as $\mathcal{W}(t) = \mathcal{U}^\dagger(t)\mathcal{W}\mathcal{U}(t)$, with

$$\mathcal{U}(t) = e^{-\frac{i}{2}A^\dagger H_t A} \ldots e^{-\frac{i}{2}A^\dagger H_1 A}. \tag{49}$$

Replacing $\mathcal{V}$ and $\mathcal{W}$ by (48) in (47), $C_1(t)$ and $C_2(t)$ become

$$\mathcal{C}_k(r,t) = \partial_w\Big|_{w=0} \partial_{v=0}\Big|_{v=0} \partial_{w'}\Big|_{w'=0} \partial_{v'}\Big|_{v'=0} \tilde{C}_k(r,t) \qquad \text{with} \qquad k = 1, 2, \tag{50}$$

$$\tilde{C}_1(r,t) = \frac{1}{2^L} \operatorname{tr}\left[\mathcal{U}^\dagger(t) e^{w+w'A^\dagger W A} \mathcal{U}(t) e^{v+v'A^\dagger W A}\right], \tag{51}$$

$$\tilde{C}_2(r,t) = \frac{1}{2^L} \operatorname{tr}\left[\mathcal{U}^\dagger(t) e^{wA^\dagger W A} \mathcal{U}(t) e^{vA^\dagger V A} \mathcal{U}^\dagger(t) e^{w'A^\dagger W A} \mathcal{U}(t) e^{v'A^\dagger V A}\right], \tag{52}$$

where $\rho = \rho_\infty = 1/2^L$. These expressions can be simplified using the equalities

$$e^{-\frac{1}{2}A^\dagger B_n A} \ldots e^{-\frac{1}{2}A^\dagger B_1 A} = e^{-\frac{1}{2}A^\dagger \tilde{B} A} \qquad \text{with} \qquad e^{\tilde{B}} = e^{B_1} \ldots e^{B_n}, \tag{53}$$

$$\operatorname{tr}\left[e^{-\frac{1}{2}A^\dagger \tilde{B} A}\right] = \left[\det\left(1 + e^{\tilde{B}}\right)\right]^{\frac{1}{2}}, \tag{54}$$

$$\det(A) = \exp(\operatorname{tr}[\log A]), \tag{55}$$

with the second one being valid for $\tilde{B}$ particle-hole symmetric (3). Applying these three expression to (51) and (52), we get

$$\tilde{C}_k(r,t) = \frac{1}{2^L} \operatorname{tr}\left[e^{-\frac{1}{2}A^\dagger \tilde{B}_k A}\right] = \frac{1}{2^L}\left[\det(1 + e^{\tilde{B}_k})\right]^{\frac{1}{2}} = \frac{1}{2^L}\exp\left(\frac{1}{2}\operatorname{tr}\left[\log(1 + e^{\tilde{B}_k})\right]\right), \tag{56}$$

where $k = 1, 2$ and

$$e^{\tilde{B}_1} \equiv e^{(v+v')V} u_t^\dagger e^{(w+w')W} u_t, \tag{57}$$

$$e^{\tilde{B}_2} \equiv e^{v'V} u_t^\dagger \ e^{w'W} u_t e^{vV} u_t^\dagger e^{wW} u_t, \tag{58}$$

$$u_t \equiv e^{-iH_t} \ldots e^{-iH_1}. \tag{59}$$

Finally, to obtain $\mathcal{C}(r,t)$ as a simple function of $W(t)$ and $V$, we must perform the derivatives in (50), making use of (56). After doing so, and using $\operatorname{tr}\mathcal{V} = \operatorname{tr}\mathcal{W} = 0$ (which greatly simplifies the result), we obtain

$$\mathcal{C}_1(r,t) = \frac{1}{2}\operatorname{tr}[W(t)V]^2 + \frac{1}{4}\operatorname{tr}[W^2(t)]\operatorname{tr}[V^2] - \operatorname{tr}[W(t)VW(t)V], \tag{60}$$

$$\mathcal{C}_2(r,t) = \frac{1}{2}\operatorname{tr}[W(t)V]^2 + \frac{1}{4}\operatorname{tr}[W^2(t)]\operatorname{tr}[V^2] + \operatorname{tr}[W(t)VW(t)V] - 2\operatorname{tr}[W^2(t)V^2]. \tag{61}$$

Combining (60) and (61) in (47) we obtain

$$\mathcal{C}(r,t) = 2\Big[C_1(r,t) - C_2(r,t)\Big], \tag{62}$$

with the new TOC and OTOC being, respectively,

$$C_1(r,t) = \operatorname{tr}[W^2(t)V^2], \tag{63}$$

$$C_2(r,t) = \operatorname{tr}[W(t)VW(t)V]. \tag{64}$$

These expressions are very similar to $\mathcal{C}_1(r,t)$ and $\mathcal{C}_2(r,t)$, with the many-body observables being replaced by the respective single-body ones. Note, however, that the correspondence between the many and single-body TOC and OTOC, given in (60) and (61), is not one-to-one. The relations (60) and (61) can be simplified if we specify the observables to be the

symmetrized particle number operator introduced below, i.e. $W = O_0$ and $V = O_r$. Since $O_r$ respects PH symmetry, one can show that $1/2 \operatorname{tr}[O_0(t)O_r]^2 = \operatorname{tr}[O_0(t)O_r O_0(t)O_r]$ and $\operatorname{tr}[O_0^2(t)]\operatorname{tr}[O_r^2] = 4$ such that

$$\mathcal{C}_1(r,t) = 1, \tag{65}$$
$$\mathcal{C}_2(r,t) = 1 + 2C_2(r,t) - 2C_1(r,t). \tag{66}$$

Knowing that the TOC and OTOC are given by (63) and (64) in terms of single-particle observables, we devote the next sections to computing these. Before proceeding to do this, we address some useful topics. Namely, we introduce the observables to be used and some convenient notation for the single-particle states.

**Symmetrized particle number observable** The ballistic or diffusive nature of the OTOC should not depend on the particular local observable used. For convenience, we consider the symmetrized particle number operator at position $r$, i.e.

$$\mathcal{O}_r(0) = \mathcal{O}_r = A^\dagger O_r A = a_r^\dagger a_r - a_r a_r^\dagger = 2a_r^\dagger a_r - 1$$
$$\Leftrightarrow O_r(0) = O_r = \operatorname{diag}(\ldots, 1_r, \ldots, -1_{r+L}, \ldots). \tag{67}$$

Specifically, $\mathcal{V}$ and $\mathcal{W}$ are the symmetrized number operators initially localized at positions $r$ and $0$, respectively: $\mathcal{V} = \mathcal{O}_r \Leftrightarrow V = O_r$ and $\mathcal{W} = \mathcal{O}_0 \Leftrightarrow W = O_0$. All the results in Section 5 are obtained for these observables.

To compare the single-body results directly to the many-body results, we simply notice that Jordan-Wigner transforming $\mathcal{O}_r$ in (67) gives the Pauli operator $Z$ acting on site $r$ used in the many-body picture calculation.

**Single-particle states** Rewriting the observables according to Nambu's notation as $\mathcal{O} = 1/2 A^\dagger O A$ allows us to work with the single-body observable $O$ in the $2L$-dimensional single-body basis. A single-particle state corresponds to a site $r$ being occupied or not, i.e. to having either a particle or a hole identified, respectively, by indices $p$ and $h$: $|r,p\rangle = |1_r\rangle$ or $|r,h\rangle = |0_r\rangle$. It is convenient to label these states by

$$|\alpha\rangle = |r_\alpha, s_\alpha\rangle = |2R_{r_\alpha} + b_{r_\alpha}, s_\alpha\rangle, \tag{68}$$

where $r_\alpha = 0, ..., L-1$ runs over the lattice sites and $s_\alpha \in \{p, h\}$ is the particle-hole index. In the second equality the position index $r_\alpha$ is decomposed as $r_\alpha = 2R_{r_\alpha} + b_{r_\alpha}$, where $R_{r_\alpha} = 0, ...., L/2-1$ labels the pair of sites $(r_\alpha, r_\alpha + 1)$ and $b_{r_\alpha} \in \{0, 1\}$ labels the position within the pair $R_{r_\alpha}$. This decomposition will prove useful since it emphasizes the structure of the circuit built with two-site unitaries. Besides, we use $|\bar{\alpha}\rangle$ to refer to $|r_\alpha, \bar{s}_\alpha\rangle$ where $\bar{p} = h$ and $\bar{h} = p$. For convenience, we will use the different notations interchangeably.

## 5.2 Moments of Haar-distributed orthogonal and unitary matrices

Consider gates on a group $\mathsf{M}(N)$ (e.g. $\mathsf{U}(N)$ or $\mathsf{O}(N)$), distributed according to the Haar measure on that group. This is the unique measure invariant under group multiplication, weighting different regions of the probability space equally and thus behaving like a uniform distribution [40]. The average with respect to the Haar probability measure $\mu$ on the matrix probability state $\mathsf{M}(N)$ is denoted by $\int_{\mathsf{M}(N)} \ldots d\mu(M)$ and abbreviated as a line over the averaged quantity. We will address to the average of products of entries of some Haar-distributed gate $u$ as moments or matrix integrals of $u$.

Since non-conserving free fermion gates are obtained from real orthogonal matrices $O$ according to $u = VOV^\dagger$, their moments can be obtained from those of Haar-distributed

orthogonal matrices. The average of products of matrix elements of a single orthogonal matrix $O$ with respect to the Haar probability measure $\mu$ on $\mathsf{O}(N)$ is in Ref. [50]. In particular, the products of two and four entries are guven by:

$$\overline{\langle\alpha_1|\,O\,|\beta_1\rangle\,\langle\alpha_2|\,O\,|\beta_2\rangle} = \frac{1}{N}\delta_{\alpha_1\alpha_2}\delta_{\beta_1\beta_2}, \tag{69}$$

$$\overline{\langle\beta_1|\,O\,|\alpha_1\rangle\,\langle\beta_2|\,O\,|\alpha_2\rangle\,\langle\beta_3|\,O\,|\alpha_3\rangle\,\langle\beta_4|\,O\,|\alpha_4\rangle}$$
$$= \frac{1}{N(N-1)(N+2)}\Big\{\delta_{\beta_1\beta_4}\delta_{\beta_2\beta_3}\Big[(N+1)\delta_{\alpha_1\alpha_4}\delta_{\alpha_2\alpha_3} - \delta_{\alpha_1\alpha_2}\delta_{\alpha_3\alpha_4} - \delta_{\alpha_1\alpha_3}\delta_{\alpha_2\alpha_4}\Big]$$
$$+ \delta_{\beta_1\beta_2}\delta_{\beta_3\beta_4}\Big[-\delta_{\alpha_1\alpha_4}\delta_{\alpha_2\alpha_3} + (N+1)\delta_{\alpha_1\alpha_2}\delta_{\alpha_3\alpha_4} - \delta_{\alpha_1\alpha_3}\delta_{\alpha_2\alpha_4}\Big]$$
$$+ \delta_{\beta_1\beta_3}\delta_{\beta_2\beta_4}\Big[-\delta_{\alpha_1\alpha_4}\delta_{\alpha_2\alpha_3} - \delta_{\alpha_1\alpha_2}\delta_{\alpha_3\alpha_4} + (N+1)\delta_{\alpha_1\alpha_3}\delta_{\alpha_2\alpha_4}\Big]\Big\}, . \tag{70}$$

Equivalent expressions for free fermion gates obtained by doing $u = VOV^\dagger$ (13) are given by (79), (101) and (102). These are very similar to the ones above, with the difference that indices of the type $\bar{\alpha}$ appear.

For the particle conserving case, the average over free fermion gates $u = \mathrm{diag}(w, w)$, where $w$ are unitary gates, can be obtained from the moments of unitary matrices, which are given by the Corollary 2.4 in [50]. In particular, the second moment of a $N \times N$ unitary matrix $w$, $\overline{\langle\alpha_1|\,w\,|\beta_1\rangle^*\,\langle\alpha_2|\,w\,|\beta_2\rangle}$, is equal to (69) while the fourth moment is given by (150).

## 5.3 Two-point correlator

Consider the two-point correlator $\mathrm{tr}[\overline{O_0(t)O_x}]$ with $O_x$ the symmetrized particle number operator. Its dynamics is determined by $O_0(t+1/2) = U^\dagger O_0(t)U$, where $U$ alternates between $U_{\mathrm{even}}$ and $U_{\mathrm{odd}}$ (11). This can be averaged considering

$$\overline{\langle\alpha_1|\,u\,|\beta_1\rangle^*\,\langle\alpha_2|\,u\,|\beta_2\rangle} = 1/N\delta_{\alpha_1\alpha_2}\delta_{\beta_1\beta_2}, \tag{71}$$

becoming

$$\overline{O_0(t+1/2)} = \sum_R\sum_{r,r'\in\{2R,2R+1\}}\sum_{s,s'}|r',s'\rangle\,\langle r,s|\overline{O_0(t)}|r,s\rangle\,\langle r',s'|, \tag{72}$$

where we considered $U = U_{\mathrm{even}}$, with analogous results following for $U = U_{\mathrm{odd}}$. Since $\sum_s\langle r,s|\overline{O_0(t)}|r,s\rangle = 0$ (due to PH symmetry), it follows that $\mathrm{tr}[\overline{O_0(t)O_x}] = 0$. Thus, to access nontrivial behavior it is necessary to probe higher order correlators.

## 5.4 NC-ST: averaged TOC, $\overline{C_1(r,t)}$

We start by considering the spatio-temporal noisy free fermion circuit without particle conservation (NC-ST), drawn in Fig. 2 (a). We first compute $\overline{C_1(r,t)}$ and then $\overline{C_2(r,t)}$. The dynamics of the later will prove to be more intricate and so we employ a vectorized notation which helps clarifying its behaviour. For a matter of consistency, this is also applied to $\overline{C_1(r,t)}$. Accordingly, we consider $C_1(r,t)$ as the overlap of two vectorized operators

$$C_1(r,t) = \mathrm{tr}[O_0^2(t)O_r^2] = \left\langle\!\left\langle O_r^2\,\middle\|\,O_0^2(t)\right\rangle\!\right\rangle, \tag{73}$$

with $||O_r^2(t)\rangle\rangle$ given by

$$||O_r^2(t)\rangle\rangle = \sum_{x,x'=0}^{L-1} \sum_{s,s'} ||x,s;x',s'\rangle\rangle \langle x,s|O_r^2(t)|x',s'\rangle, \tag{74}$$

where $||x,s;x',s'\rangle\rangle = |x,s\rangle \otimes \langle x',s'|^T$ and the particle-hole index $s$ sums over $p$ and $h$ (we usually omit these). Also, with $O_r$ the symmetrized number operator at position $r$ given by (67) we have

$$||O_r^2\rangle\rangle = \sum_s ||r,s;r,s\rangle\rangle. \tag{75}$$

Having established this vectorized formalism as the framework to use, we proceed to average $C_1(r,t)$ over multiple realizations of the random circuit and we evaluate the ensuing dynamics.

**Time evolution of $||\overline{O_0^2(t)}\rangle\rangle$**

The dynamics of $C_1(r,t)$ is completely contained in $||O_0^2(t)\rangle\rangle$. The operator $O_0^2(t)$ evolves as $O_0^2(t+1) = U^\dagger O_0^2(t) U$ which, in the vectorized notation, translates to

$$||O_0^2(t+1)\rangle\rangle = U^\dagger \otimes U^T ||O_0^2(t)\rangle\rangle. \tag{76}$$

Each circuit is a succession of layers of two-site gates acting on even and odd pairs of sites. In one unit of time one even and one odd layers are applied, $U = U_{\text{odd}} U_{\text{even}}$, such that

$$\begin{aligned}
||O_0^2(t+1)\rangle\rangle &= \left(U^\dagger \otimes U^T\right)_{\text{even}} \left(U^\dagger \otimes U^T\right)_{\text{odd}} ||O_0^2(t)\rangle\rangle \\
&= \left(\sum_{R,R'=0}^{L/2-1} u_{2R,2R+1}^\dagger \otimes u_{2R',2R'+1}^T\right) \left(\sum_{R,R'=0}^{L/2-1} u_{2R-1,2R}^\dagger \otimes u_{2R'-1,2R'}^T\right) ||O_0^2(t)\rangle\rangle,
\end{aligned} \tag{77}$$

where the sites are labelled using the pair indices and where we used (11).

**Average over disorder realizations**  Next, we average random realizations of the circuit. The average $||O_0^2(t)\rangle\rangle$ reduces to the average over the two-site gates composing the circuit. Since these are randomly chosen, they are uncorrelated and we can perform the average independently at different half-time steps. Thus, we need

$$\overline{u_{r,r+1}^\dagger \otimes u_{r',r'+1}^T} = ||\alpha_1\alpha_2\rangle\rangle \overline{\langle\beta_1|u_{r,r+1}|\alpha_1\rangle^* \langle\beta_2|u_{r',r'+1}|\alpha_2\rangle} ||\beta_1\beta_2\rangle\rangle. \tag{78}$$

This demands that we compute the average of products of entries of some Haar-distributed free fermion gate $u$. A short discussion on the topic was presented in Section 5.2. Summarizing, only moments of an even number of entries of $u$ are non-zero. This implies $\overline{u_{r,r+1}^\dagger \otimes u_{r',r'+1}^T} = \delta_{r,r'} \overline{u_{r,r+1}^\dagger \otimes u_{r,r+1}^T}$. Considering that the second moment of some free fermion gate $u$ is given by (see Section 5.2)

$$\overline{\langle\alpha_1|u|\beta_1\rangle^* \langle\alpha_2|u|\beta_2\rangle} = \frac{1}{N}\delta_{\alpha_1\alpha_2}\delta_{\beta_1\beta_2}, \tag{79}$$

where $N = \text{rank}(u) = 4$, we get

$$\overline{u_{r,r+1}^\dagger \otimes u_{r,r+1}^T} = \frac{1}{N} ||\alpha\alpha\rangle\rangle \langle\langle\beta\beta|| = ||\phi_r\rangle\rangle \langle\langle\phi_r||, \tag{80}$$

This is a projector evolving the pair of sites $(r, r+1)$ from $t$ to $t+1/2$, with

$$||\phi_r\rangle\rangle = \frac{1}{2} \sum_{x \in \{r, r+1\}} \sum_s ||x, s; x, s\rangle\rangle, \tag{81}$$

which obeys $\langle\langle\phi_r ||\phi_{r'}\rangle\rangle = \delta_{r',r} + \frac{1}{2}\delta_{r',r\pm1}$. Also, note that $||\phi_r\rangle\rangle \langle\langle\phi_r||$ leads $\overline{O_0^2(t)}$ to be always diagonal. Plugging these results into the averaged (77) leads to

$$||\overline{O_0^2(t+1)}\rangle\rangle = \left( \sum_{R=0}^{L/2-1} ||\phi_{2R}\rangle\rangle \langle\langle\phi_{2R}|| \right) \left( \sum_{R=0}^{L/2-1} ||\phi_{2R-1}\rangle\rangle \langle\langle\phi_{2R-1}|| \right) ||\overline{O_0^2(t)}\rangle\rangle. \tag{82}$$

To know $||\overline{O_0^2(t)}\rangle\rangle$ at any $t$ we start from the initial condition $||\overline{O_0^2(0)}\rangle\rangle = \sum_s ||0, s; 0, s\rangle\rangle$ and apply the above expression successively.

**Dynamics of** $||\overline{O_0^2(t)}\rangle\rangle$  Starting with $||\overline{O_0^2(0)}\rangle\rangle = \sum_s ||0, s; 0, s\rangle\rangle$ and applying the first odd layer of the circuit we obtain

$$||\overline{O_0^2(t=1/2)}\rangle\rangle = \left( \sum_{R=0}^{L/2-1} ||\phi_{2R-1}\rangle\rangle \langle\langle\phi_{2R-1}|| \right) ||\overline{O_0^2(0)}\rangle\rangle$$
$$= ||\phi_{-1}\rangle\rangle \left\langle\langle\phi_{-1} \left|\left|\overline{O_0^2(0)}\right\rangle\right\rangle\right. = ||\phi_{-1}\rangle\rangle, \tag{83}$$

to which we apply a second even layer to obtain

$$||\overline{O_0^2(t=1)}\rangle\rangle = \left( \sum_{R=0}^{L/2-1} ||\phi_{2R}\rangle\rangle \langle\langle\phi_{2R}|| \right) ||\overline{O_0^2(1/2)}\rangle\rangle$$
$$= \left( ||\phi_{-2}\rangle\rangle \langle\langle\phi_{-2}|| + ||\phi_0\rangle\rangle \langle\langle\phi_0|| \right) ||\phi_{-1}\rangle\rangle = \frac{1}{2}\left( ||\phi_{-2}\rangle\rangle + ||\phi_0\rangle\rangle \right). \tag{84}$$

Mixing neighboring pairs in successive layers is mediated by

$$||\phi_r\rangle\rangle \langle\langle\phi_r ||\phi_{r\pm1}\rangle\rangle = \frac{1}{2} ||\phi_r\rangle\rangle, \tag{85}$$

which guarantees that the subspace $\{||\phi_r\rangle\rangle\}$ is closed under time evolution. This, allied with (84), allows the decomposition

$$||\overline{O_0^2(t)}\rangle\rangle = \begin{cases} \sum_{R=0}^{L/2-1} ||\phi_{2R}\rangle\rangle \left\langle\langle\phi_{2R} \left|\left|\overline{O_0^2(t)}\right\rangle\right\rangle\right. & \text{, for } t \text{ integer} \\ \sum_{R=0}^{L/2-1} ||\phi_{2R-1}\rangle\rangle \left\langle\langle\phi_{2R-1} \left|\left|\overline{O_0^2(t)}\right\rangle\right\rangle\right. & \text{, for } t \text{ half-integer} \end{cases} \tag{86}$$

i.e. $\{||\phi_{2R}\rangle\rangle \mid R = 0, \dots, L/2-1\}$ and $\{||\phi_{2R-1}\rangle\rangle \mid R = 0, \dots, L/2-1\}$ for $t$ integer and half-integer, respectively, are the natural orthonormal basis for this problem. This is, they take full advantage of the structure coming from the average of gates, i.e. $\left\langle\langle r, s; r, s \left|\left|\overline{O_0^2(t)}\right\rangle\right\rangle\right. = \left\langle\langle r+1, s; r+1, s \left|\left|\overline{O_0^2(t)}\right\rangle\right\rangle\right.$, where $(r, r+1)$ forms a pair, and of the PH symmetry of $O_0(t)$, which leads to $\left\langle\langle r, p; r, p \left|\left|\overline{O_0^2(t)}\right\rangle\right\rangle\right. = \left\langle\langle r, h; r, h \left|\left|\overline{O_0^2(t)}\right\rangle\right\rangle\right.$. Having established the good basis to use, we can employ (85) to obtain, for $t \geq 1/2$,

$$\left\langle\langle\phi_r \left|\left|\overline{O_0^2(t+1/2)}\right\rangle\right\rangle\right. = \langle\langle\phi_r|| \left( ||\phi_{r-1}\rangle\rangle \langle\langle\phi_{r-1}|| + ||\phi_{r+1}\rangle\rangle \langle\langle\phi_{r+1}|| \right) ||\overline{O_0^2(t)}\rangle\rangle$$
$$= \frac{1}{2}\left( \langle\langle\phi_{r-1}|| + \langle\langle\phi_{r+1}|| \right) ||\overline{O_0^2(t)}\rangle\rangle, \tag{87}$$

i.e. the time-evolution of $\left\langle \langle \phi_r \left\| \overline{O_0^2(t)} \right\rangle \right\rangle$ can be understood as an averaging process with contributions from the nearest neighbors. Applying this twice gives

$$\left\langle \langle \phi_r \left\| \overline{O_0^2(t+1)} \right\rangle \right\rangle = \frac{1}{4} \left( \langle \langle \phi_{r-2} \| + 2 \langle \langle \phi_r \| + \langle \langle \phi_{r+2} \| \right) \| \overline{O_0^2(t)} \rangle \right). \tag{88}$$

This recursive expression associated to $\| \overline{O_0^2(t)} \rangle \rangle = \sum_{R=0}^{L/2-1} \| \phi_{2R} \rangle \rangle \left\langle \langle \phi_{2R} \left\| \overline{O_0^2(t)} \right\rangle \right\rangle$ is equivalent to (82), but more transparent than the latter.

**Exact result for $\overline{C_1(r,t)}$**

Now that we have analyzed the dynamics of $\| \overline{O_0^2(t)} \rangle \rangle$, we focus back on the TOC.

For $t = 0$, we use the initial condition $\| \overline{O_0^2(0)} \rangle \rangle = \sum_s \| 0, s; 0, s \rangle \rangle$ in (73) to obtain

$$\overline{C_1(r,0)} = C_1(r,0) = \left\langle \left\langle O_r^2 \left\| O_0^2(0) \right\rangle \right\rangle = 2\delta_{r,0}. \tag{89}$$

For $t \geq 1$, we decompose $\| \overline{O_0^2(t)} \rangle \rangle$ using (86) to obtain

$$\overline{C_1(r,t)} = \left\langle \left\langle O_r^2 \left\| \overline{O_0^2(t)} \right\rangle \right\rangle = \langle \langle O_r^2 \| \sum_{R=0}^{L/2-1} \| \phi_{2R} \rangle \rangle \left\langle \langle \phi_{2R} \left\| \overline{O_0^2(t)} \right\rangle \right\rangle$$

$$= \sum_{R=0}^{L/2-1} (\delta_{r,2R} + \delta_{r,2R+1}) \left\langle \langle \phi_{2R} \left\| \overline{O_0^2(t)} \right\rangle \right\rangle, \tag{90}$$

i.e. $\overline{C_1(2R,t)} = \overline{C_1(2R+1,t)} = \left\langle \langle \phi_{2R} \left\| \overline{O_0^2(t)} \right\rangle \right\rangle$. The dynamics of $\overline{C_1(r,t)}$ is then determined by that of $\left\langle \langle \phi_{2R} \left\| \overline{O_0^2(t)} \right\rangle \right\rangle$, which we saw is given by (88), starting with (84) as the initial condition, i.e.

$$\overline{C_1(2R,t+1)} = \overline{C_1(2R+1,t+1)}$$

$$= \begin{cases} \frac{1}{2}(\delta_{R-1,0} + \delta_{R,0}) & , \text{ for } t = 0 \\ \frac{1}{4} \left( \overline{C_1(2R-2,t)} + 2\overline{C_1(2R,t)} + \overline{C_1(2R+2,t)} \right) & , \text{ for } t \geq 1 \end{cases}. \tag{91}$$

Having broken down the TOC given initially by (73), we end up with a very clean picture: the circuit's structure leads $\overline{C_1(r,t)}$ to depend only on the pair $R$ to which $r$ belongs and evolving it comes down to performing a weighted average as specified in (91). This averaging process is represented pictorially in Fig. 6. In the Fig. 6(c) the 'brickwall' structure of the circuit ensures that, as time evolves, $\overline{C_1(r,t)}$ spreads across the system. Although the boundary in the $r - t$ plane between the region with zero and non-zero $\overline{C_1(r,t)}$ describes a light cone, the weights are concentrated around $r = 0$ such that the process is diffusive and not ballistic. Next, we take the continuum limit.

**Continuum limit of $\overline{C_1(r,t)}$**

Equation (91) is a discrete diffusion equation in 1D, whose continuum limit leads to the usual continuum 1D diffusion equation.

Consider the scaling form $aC_1'(x = ra, \tau = ta^2)$. For $a = 1$, this coincides with the discrete $C_1(r,t)$. If we let $a \to 0$, this approximates $C_1(r,t)$ in the continuum limit. After making this identification in (91), where $r = 2R$, we Taylor expand it up to $\mathcal{O}(a^3)$, obtaining the one-dimensional continuum diffusion equation

$$\partial_\tau \overline{C_1'(x,\tau)} = D_1 \, \partial_x^2 \overline{C_1'(x,\tau)}, \tag{92}$$

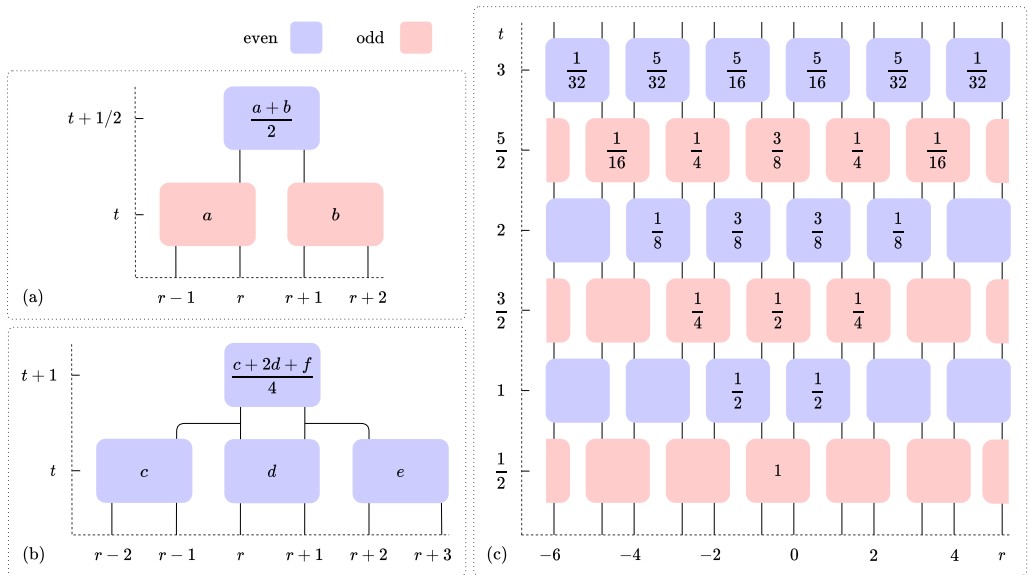

Figure 6: The values in the boxes are $\overline{C_1(r,t)}$ in the $r-t$ plane (in empty boxes $\overline{C_1(r,t)} = 0$). Panels (a) and (b) translate Eqs. (87) and (88) to a schematic, representing the evolution of $\overline{C_1(r,t)}$ in one half and one full time steps. The scheme in (b) is obtained by applying (a) twice. (c) The previous schemes can be applied to obtain $\overline{C_1(r,t)}$, starting with $C_1(r,0) = 2\delta_{r,0}$ as the initial condition. Even layers are depicted in blue while odd layers appear in red.

with diffusion coefficient $D_1 = 1$. Using the initial condition (89), which translates to $C_1'(x, t = 0) = 2\delta(x)$, the solution to (92) approximating $\overline{C_1(r,t)}$ for large $t$ and $L$ is

$$\overline{C_1(2R,t)} = \overline{C_1(2R+1,t)} \simeq a\overline{C_1'(x = 2Ra, \tau = ta^2)} = \frac{A}{\sqrt{2\pi}\sigma(t)}\exp\left(-\frac{(2R)^2}{2\sigma(t)^2}\right), \quad (93)$$

i.e. a Gaussian normalized to $A = 2$ with standard deviation $\sigma = \sqrt{2t}$.

## 5.5  NC-ST: averaged OTOC, $\overline{C_2(r,t)}$

We move on to calculate the OTOC for the non-conserving circuit with randomness in space and time (NC-ST). Although this will prove to be more intricate than $\overline{C_1(r,t)}$, the procedure follows the same steps.

We start by writing $C_2(r,t)$ as the overlap of two vectorized operators

$$C_2(r,t) = \mathrm{tr}\left[O_0(t)O_r O_0(t)O_r\right] = \langle\langle Q_r|| S ||Q_0(t)\rangle\rangle, \quad (94)$$

with $Q_r \equiv O_r \otimes O_r$ and

$$||Q_r(t)\rangle\rangle = \sum_{\alpha\beta\mu\nu} ||\alpha\beta\mu\nu\rangle\rangle \langle\alpha\beta |Q_r(t)| \mu\nu\rangle, \quad (95)$$

$$S = \sum_{\alpha\beta\mu\nu} ||\alpha\beta\mu\nu\rangle\rangle \langle\langle\alpha\beta\nu\mu||, \quad (96)$$

where $||\alpha\beta\mu\nu\rangle\rangle = |\alpha\beta\rangle \otimes \langle\mu\nu|^T$ and $S$ reorders the indices such that $\langle\langle Q_r|| S ||Q_0(t)\rangle\rangle$ is the trace present in (94). Also, with $O_r$ the symmetrized number operator at position $r$, we have

$$||Q_r\rangle\rangle = \sum_s ||rs, rs, rs, rs\rangle\rangle - ||rs, r\bar{s}, rs, r\bar{s}\rangle\rangle. \quad (97)$$

In the vectorized notation a parallel is established between $C_1(r,t) = \langle\langle O_r^2 \,||\, O_0^2(t)\rangle\rangle$ and $C_2(r,t) = \langle\langle Q_r \,||\, S \,||\, Q_0(t)\rangle\rangle$: $||O_r^2\rangle\rangle$ and $||O_0^2(t)\rangle\rangle$ are to $C_1(r,t)$ as $||Q_r\rangle\rangle$ and $||Q_0(t)\rangle\rangle$ are to $C_2(r,t)$. The dynamics of the latter is set by $||Q_0(t)\rangle\rangle$, which we proceed to evaluate.

**Time evolution of $||\overline{Q_0(t)}\rangle\rangle$**

As done for $C_1(r,t)$, we first compute how $||Q_0(t)\rangle\rangle$ evolves and then we average the result obtained.

From $O_0(t+1) = U^\dagger O_0(t) U$ it follows that $Q_0(t+1) = (U^\dagger \otimes U^\dagger) Q_0(t)(U \otimes U)$ which, in the vectorized notation, becomes

$$||Q_0(t+1)\rangle\rangle = U^\dagger \otimes U^\dagger \otimes U^T \otimes U^T \, ||Q_0(t)\rangle\rangle. \tag{98}$$

In one time step two layers of the circuit are applied, $U = U_{\text{odd}} U_{\text{even}}$. Using (11), it follows that

$$\begin{aligned}
||Q_0(t+1)\rangle\rangle &= (U^\dagger \otimes U^\dagger \otimes U^T \otimes U^T)_{\text{even}} (U^\dagger \otimes U^\dagger \otimes U^T \otimes U^T)_{\text{odd}} \, ||Q_0(t)\rangle\rangle \\
&= \left( \sum_{R,R',R'',R'''=0}^{L/2-1} u_{2R,2R+1}^\dagger \otimes u_{2R',2R'+1}^\dagger \otimes u_{2R'',2R''+1}^T \otimes u_{2R''',2R'''+1}^T \right) \\
&\quad \left( \sum_{R,R',R'',R'''=0}^{L/2-1} u_{2R-1,2R}^\dagger \otimes u_{2R'-1,2R'}^\dagger \otimes u_{2R''-1,2R''}^T \otimes u_{2R'''-1,2R'''}^T \right) ||Q_0(t)\rangle\rangle, \tag{99}
\end{aligned}$$

valid for a realization of the circuit.

**Average over disorder realizations**   We move on to average (99) over random realizations of the circuit. As before, we take the average of each layer independently from the others. Thus, it suffices to compute

$$\begin{aligned}
\overline{u_{r,r+1}^\dagger \otimes u_{r',r'+1}^\dagger \otimes u_{r'',r''+1}^T \otimes u_{r''',r'''+1}^T} &= ||\alpha_1 \alpha_2 \alpha_3 \alpha_4\rangle\rangle \langle\langle \beta_1 \beta_2 \beta_3 \beta_4|| \cdot \\
&\quad \cdot \overline{\langle\beta_1|u_{r,r+1}|\alpha_1\rangle^* \langle\beta_2|u_{r',r'+1}|\alpha_2\rangle^* \langle\beta_3|u_{r'',r''+1}|\alpha_3\rangle \langle\beta_4|u_{r''',r'''+1}|\alpha_4\rangle}. \tag{100}
\end{aligned}$$

To perform this average we refer again to Section 5.2. Recall that only even moments of $u$ are non-zero. The second moments are given either by (79) and

$$\overline{\langle\beta_1|\, u \,|\alpha_1\rangle \langle\beta_2|\, u \,|\alpha_2\rangle} = \overline{\langle\beta_1|\, u \,|\alpha_1\rangle^* \langle\beta_2|\, u \,|\alpha_2\rangle^*} = \frac{1}{N} \delta_{\bar\alpha_1 \alpha_2} \delta_{\bar\beta_1 \beta_2}, \tag{101}$$

and the fourth moment is given by

$$\begin{aligned}
&\overline{\langle\beta_1|\, u \,|\alpha_1\rangle^* \langle\beta_2|\, u \,|\alpha_2\rangle^* \langle\beta_3|\, u \,|\alpha_3\rangle \langle\beta_4|\, u \,|\alpha_4\rangle} \\
&= \frac{1}{N(N-1)(N+2)} \Big\{ \delta_{\beta_1\beta_4}\delta_{\beta_2\beta_3}\big[(N+1)\delta_{\alpha_1\alpha_4}\delta_{\alpha_2\alpha_3} - \delta_{\bar\alpha_1\alpha_2}\delta_{\bar\alpha_3\alpha_4} - \delta_{\alpha_1\alpha_3}\delta_{\alpha_2\alpha_4}\big] \\
&\quad + \delta_{\bar\beta_1\beta_2}\delta_{\bar\beta_3\beta_4}\big[-\delta_{\alpha_1\alpha_4}\delta_{\alpha_2\alpha_3} + (N+1)\delta_{\bar\alpha_1\alpha_2}\delta_{\bar\alpha_3\alpha_4} - \delta_{\alpha_1\alpha_3}\delta_{\alpha_2\alpha_4}\big] \\
&\quad + \delta_{\beta_1\beta_3}\delta_{\beta_2\beta_4}\big[-\delta_{\alpha_1\alpha_4}\delta_{\alpha_2\alpha_3} - \delta_{\bar\alpha_1\alpha_2}\delta_{\bar\alpha_3\alpha_4} + (N+1)\delta_{\alpha_1\alpha_3}\delta_{\alpha_2\alpha_4}\big] \Big\}, \tag{102}
\end{aligned}$$

with $N = \text{rank}(u) = 4$. This means that, when performing the average in (100), either two or four entries of $u$ must refer to the same unitary, leaving us with four possible terms. The case where all entries refer to the same unitary (i.e. $r = r' = r'' = r'''$) contributes with

one term while the other three come from two different unitaries appearing, for which there are three configurations corresponding to the pairings we can make out of four elements: (12)(34), (13)(24) and (14)(23). Taking (79), (101) and (102) into account, the sum over (100) becomes

$$\sum_{r,r',r'',r'''=0}^{L/2-1} \overline{u^\dagger_{r,r+1} \otimes u^\dagger_{r',r'+1} \otimes u^T_{r'',r''+1} \otimes u^T_{r''',r'''+1}} = \sum_{r,r'=0}^{L/2-1} v_{r,r'}, \tag{103}$$

with

$$v_{r,r'} = (1 - \delta_{rr'}) \Big( ||\Theta\rangle\rangle \langle\langle\Theta|| + ||\bar{\Theta}\rangle\rangle \langle\langle\bar{\Theta}|| + ||\tilde{\Theta}\rangle\rangle \langle\langle\tilde{\Theta}|| \Big)_{r,r'}$$

$$+ \frac{\delta_{rr'}}{(N+2)} \Big( \Big[ (N+1) ||\Theta\rangle\rangle - ||\bar{\Theta}\rangle\rangle - ||\tilde{\Theta}\rangle\rangle \Big] \langle\langle\Theta||$$

$$+ \Big[ - ||\Theta\rangle\rangle + (N+1) ||\bar{\Theta}\rangle\rangle - ||\tilde{\Theta}\rangle\rangle \Big] \langle\langle\bar{\Theta}|| + \Big[ - ||\Theta\rangle\rangle - ||\bar{\Theta}\rangle\rangle + (N+1) ||\tilde{\Theta}\rangle\rangle \Big] \langle\langle\tilde{\Theta}|| \Big)_{r,r}, \tag{104}$$

where we use the definitions

$$||\Theta_{r,r'}\rangle\rangle = \frac{1}{\sqrt{g_{r,r'}}} \sum_{\substack{r_\alpha\in\{r,r+1\} \\ r_\beta\in\{r',r'+1\}}} \sum_{s_\alpha,s_\beta} ||\alpha\beta\beta\alpha\rangle\rangle, \tag{105}$$

$$||\bar{\Theta}_{r,r'}\rangle\rangle = \frac{1}{\sqrt{g_{r,r'}}} \sum_{\substack{r_\alpha\in\{r,r+1\} \\ r_\beta\in\{r',r'+1\}}} \sum_{s_\alpha,s_\beta} ||\alpha\bar{\alpha}\beta\bar{\beta}\rangle\rangle, \tag{106}$$

$$||\tilde{\Theta}_{r,r'}\rangle\rangle = \frac{1}{\sqrt{g_{r,r'}}} \sum_{\substack{r_\alpha\in\{r,r+1\} \\ r_\beta\in\{r',r'+1\}}} \sum_{s_\alpha,s_\beta} ||\alpha\beta\alpha\beta\rangle\rangle, \tag{107}$$

with $g_{r,r'} = N(N - \delta_{r,r'})$. Since $r$ and $r'$ have the same parity, these obey

$$\langle\langle i_{rr'} ||i_{qq'}\rangle\rangle = \frac{16}{g_{r,r'}}\delta_{q,r}\delta_{q',r'} + \frac{4}{\sqrt{g_{rr'}g_{qq'}}}\delta_{q,r\pm1}\delta_{q',r'\pm1}, \tag{108}$$

$$\langle\langle i_{rr'} ||j_{qq'}\rangle\rangle = \frac{\delta_{r,r'}}{6}\Big( 2\delta_{q,r}\delta_{q',r'} + \delta_{q,r+1}\delta_{q',r'+1} + \delta_{q,r-1}\delta_{q',r'-1} \Big), \tag{109}$$

for $i \neq j \in \{\Theta, \bar{\Theta}, \tilde{\Theta}\}$. Notice that, while the evolution of $||\overline{O_0^2(t)}\rangle\rangle$ in $C_1(r,t)$ involved mixing within each pair of sites $(r, r+1)$ through the projector $||\phi_r\rangle\rangle \langle\langle\phi_r||$ (80), here the evolution through $v_{r,r'}$ entails mixing between each two pairs $(r, r+1)$ and $(r', r'+1)$. Thus, we expect the time evolution of $||\overline{O_0^2(t)}\rangle\rangle$ to be a two-dimensional process in space.

Finally, considering (103), the average of (99) becomes

$$||\overline{Q_0(t+1)}\rangle\rangle = \left( \sum_{R,R'=0}^{L/2-1} v_{2R,2R'} \right) \left( \sum_{R,R'=0}^{L/2-1} v_{2R-1,2R'-1} \right) ||\overline{Q_0(t)}\rangle\rangle, \tag{110}$$

with $v_{r,r'}$ given by (104). Starting with $||\overline{Q_0(0)}\rangle\rangle$ given by (97), applying this expression recursively gives $||\overline{Q_0(t)}\rangle\rangle$ at successive time steps. Next, we do this and simplify (110).

**Dynamics of $||\overline{Q_0(t)}\rangle\rangle$**    We start by defining

$$||\Theta\bar{\Theta}_{r,r'}\rangle\rangle = ||\Theta_{r,r'}\rangle\rangle - ||\bar{\Theta}_{r,r'}\rangle\rangle. \tag{111}$$

Starting with $||\overline{Q_0(0)}\rangle\rangle$ given by (97), applying the first odd layer of the circuit leads to

$$||\overline{Q_0(t=1/2)}\rangle\rangle = \left(\sum_{R,R'=0}^{L/2-1} v_{2R-1,2R'-1}\right)||Q_0(0)\rangle\rangle = v_{-1,-1}||Q_0(0)\rangle\rangle = \frac{1}{\sqrt{3}}||\Theta\bar{\Theta}_{-1,-1}\rangle\rangle, \tag{112}$$

to which we apply a second even layer to obtain

$$||\overline{Q_0(t=1)}\rangle\rangle = \left(\sum_{R,R'=0}^{L/2-1} v_{2R,2R'}\right)||\overline{Q_0(1/2)}\rangle\rangle = (v_{-2,-2} + v_{-2,0} + v_{0,-2} + v_{0,0})||\overline{Q_0(1/2)}\rangle\rangle$$

$$= \frac{1}{6\sqrt{3}}\left(||\Theta\bar{\Theta}_{-2,-2}\rangle\rangle + ||\Theta\bar{\Theta}_{0,0}\rangle\rangle\right) + \frac{1}{6}\left(||\Theta\bar{\Theta}_{-2,0}\rangle\rangle + ||\Theta\bar{\Theta}_{0,-2}\rangle\rangle\right). \tag{113}$$

Also, we use (108) and (109) to see that $||\Theta\bar{\Theta}_{r,r'}\rangle\rangle$ evolves under $v_{r,r'}$ according to

$$v_{r+a,r'+a}||\Theta\bar{\Theta}_{r,r'}\rangle\rangle = \left(\frac{1}{4} - \frac{\delta_{r,r'}}{12}\right)||\Theta\bar{\Theta}_{r+a,r'+a}\rangle\rangle, \tag{114}$$

$$v_{r+a,r'-a}||\Theta\bar{\Theta}_{r,r'}\rangle\rangle = \left[\frac{1}{4} + \left(\frac{1}{2\sqrt{3}} - \frac{1}{4}\right)(\delta_{r,r'} + \delta_{r+a,r'-a})\right]||\Theta\bar{\Theta}_{r+a,r'-a}\rangle\rangle, \tag{115}$$

where $a = \pm 1$. These relations control the mixing occurring between neighboring pairs in successive time steps and ensure that the subspace $\{||\Theta\bar{\Theta}_{r,r'}\rangle\rangle\}$ is closed under time evolution. This, allied with (112), allows the decomposition

$$||\overline{Q_0(t)}\rangle\rangle = \begin{cases} \sum_{R=0}^{L/2-1} \frac{1}{2}||\Theta\bar{\Theta}_{2R,2R'}\rangle\rangle \left\langle\left\langle\Theta\bar{\Theta}_{2R,2R'}\left|\left|\overline{Q_0(t)}\right\rangle\right\rangle\right. & \text{, for } t \text{ integer} \\ \sum_{R=0}^{L/2-1} \frac{1}{2}||\Theta\bar{\Theta}_{2R-1,2R'-1}\rangle\rangle \left\langle\left\langle\Theta\bar{\Theta}_{2R-1,2R'-1}\left|\left|\overline{Q_0(t)}\right\rangle\right\rangle\right. & \text{, for } t \text{ half-integer} \end{cases} \tag{116}$$

where the factor $1/2$ arises from $\langle\langle\Theta\bar{\Theta}_{r,r'}||\Theta\bar{\Theta}_{q,q'}\rangle\rangle = 2\delta_{r,q}\delta_{r',q'}$, where the indices are restricted to having the same parity. As $\{||\phi_r\rangle\rangle\}$ was the natural basis on which to express $||\overline{O_0^2(t)}\rangle\rangle$, so it happens that $\{\frac{1}{\sqrt{2}}||\Theta\bar{\Theta}_{2R,2R'}\rangle\rangle \mid R, R' = 0, \ldots, L/2-1\}$ and $\{\frac{1}{\sqrt{2}}||\Theta\bar{\Theta}_{2R-1,2R'-1}\rangle\rangle \mid R, R' = 0, \ldots, L/2-1\}$ are the natural orthonormal basis on which to express $||\overline{Q_0(t)}\rangle\rangle$. Let us comment on the states $||\alpha\beta\mu\nu\rangle\rangle$ grouped under the same $||\Theta\bar{\Theta}_{r,r'}\rangle\rangle$. Due to PH symmetry, out of the three types of states surviving the average of Haar-distributed gates $-$ $||\alpha\beta\beta\alpha\rangle\rangle$, $||\alpha\bar{\alpha}\beta\bar{\beta}\rangle\rangle$ and $||\alpha\beta\alpha\beta\rangle\rangle$ $-$ only the first two survive; explicitly: $\langle\langle\alpha\beta\alpha\beta||Q_0(t)\rangle\rangle = -\langle\langle\overline{\alpha}\beta\overline{\alpha}\beta||Q_0(t)\rangle\rangle \Rightarrow \sum_{s_\alpha}\langle\langle\alpha\beta\alpha\beta||Q_0(t)\rangle\rangle = 0$. Furthermore, PH symmetry implies $\langle\langle\alpha\beta\beta\alpha||Q_0(t)\rangle\rangle = -\left\langle\left\langle\alpha\bar{\alpha}\beta\bar{\beta}||Q_0(t)\right\rangle\right\rangle = |\langle\alpha|O_0(t)|\beta\rangle|^2$ such that $||\Theta\bar{\Theta}_{r,r'}\rangle\rangle$ groups the states $||\Theta_{r,r'}\rangle\rangle$ and $-||\bar{\Theta}_{r,r'}\rangle\rangle$ which evolve equally. The structure appearing restricts the dynamics of $||Q_0(t)\rangle\rangle$ such that it is simpler than one might have guessed, with only $8L^2$ entries out of $(2L)^4$ being non-zero.

Having established the good basis to use, we employ (114) and (115) to obtain, for $t \geq 1/2$,

$$\left\langle\left\langle\Theta\bar{\Theta}_{r,r'}\left|\left|\overline{Q_0(t+1/2)}\right\rangle\right\rangle\right. =$$

$$= \langle\langle\Theta\bar{\Theta}_{r,r'}||\left(v_{r-1,r'-1} + v_{r+1,r'+1} + v_{r-1,r'+1} + v_{r+1,r'-1}\right)||\overline{Q_0(t)}\rangle\rangle$$

$$= \text{tr}\left[m_{r,r'}\begin{pmatrix} \langle\langle\Theta\bar{\Theta}_{r-1,r'-1}|| & \langle\langle\Theta\bar{\Theta}_{r-1,r'+1}|| \\ \langle\langle\Theta\bar{\Theta}_{r+1,r'-1}|| & \langle\langle\Theta\bar{\Theta}_{r+1,r'+1}|| \end{pmatrix}^T ||\overline{Q_0(t)}\rangle\rangle\right], \tag{117}$$

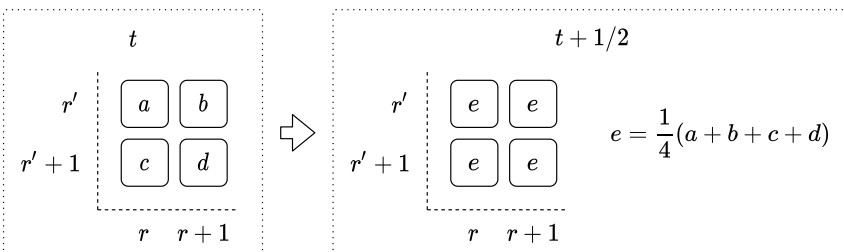

Figure 7: Scheme of the evolution of the bulk $\left\langle\left\langle\Theta\bar{\Theta}_{r,r'}\left\|\overline{Q_0(t)}\right\rangle\right\rangle|_{|r-r'|>2}$ (values in boxes) in one-half time step. The bulk evolution uses the third matrix $m_{r,r'}$ in (118) and is effectively an average of neighboring elements $\left\langle\left\langle\Theta\bar{\Theta}_{r,r'}\left\|\overline{Q_0(t)}\right\rangle\right\rangle$ whose position indices belong to the pairs $(r, r+1)$ and $(r', r'+1)$. Depending on whether $t+1/2$ is integer or half-integer, these pairs are even or odd.

where $m_{r,r'}$ is a coefficient matrix which assumes different values depending on $r$ and $r'$:

$$m_{r,r} = \begin{pmatrix} \frac{1}{6} & \frac{1}{2\sqrt{3}} \\ \frac{1}{2\sqrt{3}} & \frac{1}{6} \end{pmatrix} \quad , \quad m_{r,r+2} = m_{r,r-2}^T = \begin{pmatrix} \frac{1}{4} & \frac{1}{4} \\ \frac{1}{2\sqrt{3}} & \frac{1}{4} \end{pmatrix} \quad , \quad m_{r,r'} = \frac{1}{4}\begin{pmatrix} 1 & 1 \\ 1 & 1 \end{pmatrix}, \quad (118)$$

with the last $m_{r,r'}$ being valid for $|r - r'| > 2$. In the bulk, i.e. for $|r - r'| > 2$, this evolution equation is an average over neighboring sites, as depicted in Fig. 7. This is similar to what is shown to happen for $\overline{C_1(r,t)}$ in Fig. 6, where the process occurring is one instead of two-dimensional.

Applying (117) twice gives

$$\left\langle\left\langle\Theta\bar{\Theta}_{r,r'}\left\|\overline{Q_0(t+1)}\right\rangle\right\rangle =$$

$$= \text{tr}\left[M_{r,r'}\begin{pmatrix} \langle\langle\Theta\bar{\Theta}_{r-2,r'-2}|| & \langle\langle\Theta\bar{\Theta}_{r-2,r'}|| & \langle\langle\Theta\bar{\Theta}_{r-2,r'+2}|| \\ \langle\langle\Theta\bar{\Theta}_{r,r'-2}|| & \langle\langle\Theta\bar{\Theta}_{r,r'}|| & \langle\langle\Theta\bar{\Theta}_{r,r'+2}|| \\ \langle\langle\Theta\bar{\Theta}_{r+2,r'-2}|| & \langle\langle\Theta\bar{\Theta}_{r+2,r'}|| & \langle\langle\Theta\bar{\Theta}_{r+2,r'+2}|| \end{pmatrix}^T ||\overline{Q_0(t)}\rangle\rangle\right] \quad (119)$$

with $M_{r,r'}$ a coefficient matrix given by

$$M_{r,r} = \frac{1}{72}\begin{pmatrix} 2 & 5\sqrt{3} & 3\sqrt{3} \\ 5\sqrt{3} & 16 & 5\sqrt{3} \\ 3\sqrt{3} & 5\sqrt{3} & 2 \end{pmatrix} \quad , \quad M_{r,r+2} = M_{r,r-2}^T = \frac{1}{48}\begin{pmatrix} 3 & 6 & 3 \\ \frac{10}{\sqrt{3}} & 13 & 6 \\ 4 & \frac{10}{\sqrt{3}} & 3 \end{pmatrix}$$

$$M_{r,r+4} = M_{r,r-4}^T = \frac{1}{16}\begin{pmatrix} 1 & 2 & 1 \\ 2 & 4 & 2 \\ \frac{2}{\sqrt{3}} & 2 & 1 \end{pmatrix} \quad , \quad M_{r,r'} = \frac{1}{16}\begin{pmatrix} 1 & 2 & 1 \\ 2 & 4 & 2 \\ 1 & 2 & 1 \end{pmatrix}, \quad (120)$$

where the last $M_{r,r'}$ is valid for $|r - r'| > 4$. This last matrix establishes the evolution of the bulk of $\left\langle\left\langle\Theta\bar{\Theta}_{r,r'}\left\|\overline{Q_0(t)}\right\rangle\right\rangle$ as a weighted average resulting from applying the scheme seen in Fig. 7 twice.

At last, starting with (113) as the initial condition, $||\overline{Q_0(t)}\rangle\rangle$ is given by the recursive expression (119) alongside (116). This is equivalent to (110), only simplified. Now we are equipped to compute the OTOC.

**Exact result for $\overline{C_2(r,t)}$**

Let us obtain the final result for $\overline{C_2(r,t)}$. At $t = 0$, $||Q_0(0)\rangle\rangle$ is given by (97) such that (94) becomes

$$\overline{C_2(r,t=0)} = C_2(r,t=0) = \langle\langle Q_r || S || Q_0(0)\rangle\rangle = 2\delta_{r,0}. \tag{121}$$

For $t \geq 1$, we decompose $||\overline{Q_0(t)}\rangle\rangle$ using (116) to obtain

$$\overline{C_2(r,t)} = \langle\langle Q_r || S ||\overline{Q_0(t)}\rangle\rangle = \langle\langle Q_r || S \left( \sum_{R,R'=0}^{L/2-1} \frac{1}{2} ||\Theta\bar{\Theta}_{2R,2R'}\rangle\rangle \left\langle\langle \Theta\bar{\Theta}_{2R,2R'} \left|\left| \overline{Q_0(t)}\right.\right\rangle\right\rangle \right)$$

$$= \frac{1}{2\sqrt{3}} \sum_{R=0}^{L/2-1} (\delta_{r,2R} + \delta_{r,2R+1}) \left\langle\langle \Theta\bar{\Theta}_{2R,2R} \left|\left| \overline{Q_0(t)}\right.\right\rangle\right\rangle. \tag{122}$$

For convenience, we define

$$K_{r,r'}(t) = \frac{1}{2\sqrt{3}} \left\langle\langle \Theta\bar{\Theta}_{r,r'} \left|\left| \overline{Q_0(t)}\right.\right\rangle\right\rangle, \tag{123}$$

whose diagonal gives the OTOC: $\overline{C_2(2R,t)} = \overline{C_2(2R+1,t)} = K_{2R,2R}(t)$. Thus, we use (113) to get the OTOC at $t = 1$:

$$\overline{C_2(2R,t=1)} = \overline{C_2(2R+1,t=1)} = \frac{1}{18}\left(\delta_{R,-1} + \delta_{R,0}\right), \tag{124}$$

At subsequent time steps the dynamics of $\overline{C_2(r,t)}$ is determined by that of $\left\langle\langle \Theta\bar{\Theta}_{r,r'} \left|\left| \overline{O_0^2(t)}\right.\right\rangle\right\rangle$ $\Leftrightarrow K_{r,r'}(t)$ we saw to be given by (119), i.e. for $t \geq 1$ we have

$$\overline{C_2(2R,t+1)} = \overline{C_2(2R+1,t+1)} = \text{diag}[K_{2R,2R'}(t+1)]$$

$$= \text{diag}\left\{ \text{tr}\left[ M_{2R,2R'} \begin{pmatrix} K_{2R-2,2R'-2} & K_{2R-2,2R'} & K_{2R-2,2R'+2} \\ K_{2R,2R'-2} & K_{2R,2R'} & K_{2R,2R'+2} \\ K_{2R+2,2R'-2} & K_{2R+2,2R'} & K_{2R+2,2R'+2} \end{pmatrix}^T (t) \right] \right\} \tag{125}$$

Note that, although the OTOC is given by the diagonal of $K_{r,r'}(t)$, we must keep track of the whole matrix to learn its dynamics. Before proceeding to inspect the dynamics of $K_{r,r'}(t)$ in the continuum limit, we give a pictorial view in Fig. 8 of how $K_{r,r'}(t)$ (and thus, the OTOC) evolves in half-time steps.

**Continuum limit of $\overline{C_2(r,t)}$**

Here, we recover the 2D continuum diffusion equation by taking the continuum limit of (117), which evolves $K_{r,r'}(t)$ in half time steps. Note that this equation (or, more specifically, the matrix $m_{r,r'}$) differs depending on whether $r' = r$, $r' = r \pm 2$ or $|r'-r| > 2$. However, since in the thermodynamic limit ($L \to \infty$) the regions with $r' = r$ and $r' = r \pm 2$ constitute a set of measure zero where $K_{r,r'}(t)$ does not diverge, they are not expected to influence the behaviour of the bulk. Thus, we start by obtaining a continuum diffusion equation for the bulk, i.e. for $K_{r,r'}(t)$ in $|r'-r| > 2$. Then, wishing to obtain the OTOC, given by $K_{r,r}(t)$, we see that the diagonal differs from the bulk by a constant factor.

Consider the scaling form $a^2 K'(x = ra, x' = r'a, \tau = ta^2)$. For $a = 1$, this coincides with the discrete $K_{r,r'}(t)$. If we let $a \to 0$, this approximates $K_{r,r'}(t)$ in the continuum limit, i.e. $\lim_{L\to\infty, t\to\infty} K_{r,r'}(t) = \lim_{a\to 0} a^2 K'(ra, r'a, ta^2)$. After making this identification in (119) and considering $m_{r,r'}|_{|r-r'|>2}$ to be the isotropic bulk coefficient matrix

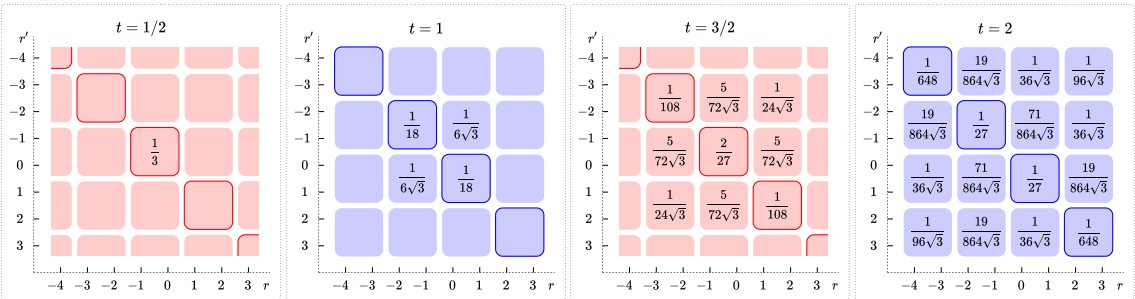

Figure 8: The values in the boxes correspond to $K_{r,r'}(t)$ in the $r - r'$ plane, shown here for $t \in \{1/2, 1, 3/2, 2\}$ (empty boxes have zero value). Starting with $K_{r,r'}(t) = 2\delta_{r,r',0}$ as the initial condition, we evolve this by applying (117) successively. The highlighted diagonal gives $\overline{C_2(r,t)}$. Note that the boxes group the pairs of sites $(r, r+1)$ and $(r', r'+1)$ enhancing the symmetry $K_{r,r'}(t) = K_{r+1,r'+1}(t) = K_{r+1,r'}(t) = K_{r,r'+1}(t)$. Depending on whether $t$ is integer or half-integer (blue/red), these pairs are even or odd.

given in (118), we Taylor expand $K'_{\text{bulk}}(x, x', \tau)$, i.e. the quantity $K'(x, x', \tau)$ evolving isotropically under (119) with $m_{r,r'}|_{|r-r'|>2}$, up to $\mathcal{O}(a^2)$ such that (117) becomes

$$\partial_\tau K'_{\text{bulk}}(x, x', \tau) = D_2(\partial_x^2 + \partial_{x'}^2) K'_{\text{bulk}}(x, x', \tau), \quad (126)$$

i.e. the isotropic diffusion equation in 2D, with diffusion constant $D_2 = 1$.

Let us address the diagonal $K_{r,r}(t)$. Namely, we account for the anisotropy present in the evolution of the diagonal which impacts the way the diagonal initial condition is distributed along different directions, something which we disregarded to obtain (126). To begin with, we distinguish between $r = r'$ and $r \neq r'$ and approximate $K_{r,r}(t)$ by $a^2 K'(x, x, \tau)$ for $r' = r$ and by $K'_{\text{bulk}}(x, x', \tau)$ for $r' \neq r$. Making this identification in (117) gives

$$\begin{aligned}
K'_{\text{bulk}}&(x - a, x + a, \tau + 1/2) \\
&= 1/4[K'_{\text{bulk}}(x - 2a, x, \tau) + K'_{\text{bulk}}(x - 2a, x + 2a, \tau) + K'_{\text{bulk}}(x, x + 2a, \tau)] \\
&\quad + 1/(2\sqrt{3}) K'(x, x, \tau)
\end{aligned} \quad (127)$$

which, after Taylor expanding $K'_{\text{bulk}}$ up to the lowest order, $\mathcal{O}(a^0)$, yields

$$K'(x, x, \tau) \approx c K'_{\text{bulk}}(x, x, \tau), \quad (128)$$

with $c = \sqrt{3}/2$. Note that we approximated $K'(x, x \pm 2a, \tau)$ by the bulk $K'_{\text{bulk}}$: similarly to what is done to obtain (128), we can Taylor expand $K'_{\text{bulk}}$ up to $\mathcal{O}(a^0)$ in the evolution equation of $K'_{\text{bulk}}(x - 2a, x + 2a, \tau + 1/2)$ to obtain $K'(x, x \pm 2a, \tau) \sim K'_{\text{bulk}}(x, x' \pm 2a, \tau)$.

Considering (128) in the evolution equation of the diagonal (117) leads to the weighted average

$$K'_{\text{bulk}}(x, x, \tau + 1/2) = \text{tr}\left[ m'_{r,r'} \begin{pmatrix} K'_{\text{bulk}}(x - 2, x - 2, \tau) & K'_{\text{bulk}}(x - 2, x + 2, \tau) \\ K'_{\text{bulk}}(x + 2, x - 2, \tau) & K'_{\text{bulk}}(x + 2, x + 2, \tau) \end{pmatrix}^T \right], \quad (129)$$

with

$$m'_{r,r} = \frac{1}{6}\begin{pmatrix} 1 & 2 \\ 2 & 1 \end{pmatrix}. \quad (130)$$

This allows us to directly compare the diagonal with the bulk. The anisotropy in the new coefficient matrix $m'_{r,r}$ leads to diffusion with weighting factor $f = 1/2$ (the anti-diagonal elements are twice the diagonal ones). Thus, to obtain $K'(x, x, \tau)$, we take the diagonal of $K'_{\text{bulk}}(x, x', \tau)$ given as the solution of (126) and multiply it by $c$ − which accounts for the different scaling of the diagonal − and by $f$ − which accounts for the anisotropy present.

Finally, we obtain the solution of (126) and, hence, the OTOC. The initial condition (121) translates to $K'(x, x, \tau = 0) \approx cK'_{\text{bulk}}(x, x, \tau = 0) = A\delta(x)\delta(x')$, with $A = 2$ and where we used (128), such that the solution to (126) is

$$cK'_{\text{bulk}}(x, x', \tau) = \frac{A}{4\pi\tau D_2} \exp\left(-\frac{x^2 + x'^2}{4D_2\tau}\right). \tag{131}$$

Finally, the OTOC is approximated by (131) times the corrective factor $f$

$$\overline{C_2(2R, t)} = \overline{C_2(2R + 1, t)} = K_{2R,2R}(t)$$

$$\simeq fca^2 K'_{\text{bulk}}(2Ra, 2Ra, \tau) = \frac{A(t)}{\sqrt{2\pi}\sigma(t)} \exp\left(-\frac{(2R)^2}{2\sigma(t)^2}\right), \tag{132}$$

i.e. in the continuum limit, the OTOC is described by a Gaussian with broadening width $\sigma(t) = \sqrt{t}$ and varying normalization $A(t) = fA/(2\sqrt{2\pi t})$, with $A = 2$ and $f = 1/2$. We conclude that $\overline{C_2(r, t)}$ is approximately the diagonal of a quantity which diffuses in 2D, analogously to $\overline{C_1(r, t)}$ which instead diffused in 1D.

## 5.6 NC-T: averaged TOC, $\overline{C_1(r, t)}$

Now, we turn to the non particle conserving case, but with randomness only in time (NC-T). This is realized by the circuit in Fig 2(b) − each layer $U_{\text{even}}$ and $U_{\text{odd}}$ is built acting with the same randomly chosen gate $u$ on all pairs of sites such that the system is invariant under space translation by multiples of two sites.

Here, we derive the behaviour of $\overline{C_1(r, t)}$ by pointing out the differences from the NC-ST case, which reside in the average of unitaries. We saw in Section 5.4 that the average of unitaries $u^\dagger_{r,r+1} \otimes u^T_{r',r'+1}$ is non-zero if $u_{r,r+1} = u_{r',r'+1}$. While in the NC-ST case this demanded $r' = r$, the spatial homogeneity of NC-T softens this condition to $r' = r$ modulo 2. As a consequence, the operator $\sum_r ||\phi_r\rangle\rangle \langle\langle\phi_r||$ which evolves $||\overline{O_0^2(t)}\rangle\rangle$ in NC-ST is generalized to $\sum_{rr'} ||\phi_{rr'}\rangle\rangle \langle\langle\phi_{rr'}||$ with

$$||\phi_r\rangle\rangle \longrightarrow ||\phi_{rr'}\rangle\rangle = \frac{1}{2} \sum_{b\in\{0,1\}} \sum_s ||r + b, s; r' + b, s\rangle\rangle. \tag{133}$$

Considering the new evolution operator, we start from $||\overline{O_0^2(0)}\rangle\rangle = \sum_s ||0, s; 0, s\rangle\rangle$ to obtain

$$||\overline{Q_0(1/2)}\rangle\rangle = \left(\sum_{R,R'=0}^{L/2-1} ||\phi_{2R,2R'}\rangle\rangle \langle\langle\phi_{2R,2R'}||\right) ||\overline{Q_0(0)}\rangle\rangle = ||\phi_{-1-1}\rangle\rangle = ||\phi_{-1}\rangle\rangle. \tag{134}$$

The interaction between neighboring pairs in successive layers analogous to (85) is

$$||\phi_{r,r'}\rangle\rangle \langle\langle\phi_{r,r'} ||\phi_{r+a,r'+a}\rangle\rangle = \frac{1}{2} ||\phi_{r,r'}\rangle\rangle, \tag{135}$$

with $a = \pm 1$. This leads to an evolution equation analogous to (87)

$$\left\langle\left\langle\phi_{r,r'} \left\|\overline{O_0^2(t + 1/2)}\right\rangle\right\rangle =\right.$$

$$= \langle\langle\phi_{r,r'}|| \left(||\phi_{r-1,r'-1}\rangle\rangle \langle\langle\phi_{r-1,r'-1}|| + ||\phi_{r+1,r'+1}\rangle\rangle \langle\langle\phi_{r+1,r'+1}||\right) ||\overline{O_0^2(t)}\rangle\rangle$$

$$= \frac{1}{2}\left(\langle\langle\phi_{r-1,r'-1}|| + \langle\langle\phi_{r+1,r'+1}||\right) ||\overline{O_0^2(t)}\rangle\rangle. \tag{136}$$

This expression alongside the initial condition (134) leads only terms of the type $||\phi_{r,r}\rangle\rangle = ||\phi_r\rangle\rangle$ to arise. Thus, the above expression is effectively reduced to (87). Since the initial condition $||\overline{Q_0(1/2)}\rangle\rangle) = ||\phi_{-1}\rangle\rangle$ is the same we had for NC-ST, $\overline{C_1(r,t)}$ in the NC-T case to be exactly equal to the one obtained for NC-ST. Although the spatial homogeneity of the new circuit threatens the TOC dynamics to be described by a two-dimensional process, the initial condition fixes it to the one-dimensional process already known for NC-ST.

## 5.7   NC-T: averaged OTOC, $\overline{C_2(r,t)}$

We now compute $\overline{C_2(r,t)}$ for the NC-T case. As it happens with $\overline{C_1(r,t)}$, the difference from the NC-ST case lies in the average of unitaries.

   We saw in Section 5.5 that, for the average $\overline{u_{r_1,r_1+1}^\dagger \otimes u_{r_2,r_2+1}^\dagger \otimes u_{r_1',r_1'+1}^T \otimes u_{r_2',r_2'+1}^T}$ to be non-zero,, either two or four unitaries must be equal, which, in the NC-ST case, translates to $r_1 = r_2 \wedge r_1' = r_2'$ or $r_1 = r_1' \wedge r_2 = r_2'$ or $r_1 = r_2' \wedge r_2 = r_1'$ or $r_1 = r_2 = r_1' = r_2'$. Similarly to what we just saw for $\overline{C_1(r,t)}$, these restrictions are softened in the NC-T case: since all the gates composing one layer are the same the four unitaries can refer to different pairs such that $v_{r,r'}$ is generalized to $v_{r_1,r_2,r_1',r_2'}$ by transforming (105), (106) and (107) according to

$$
\begin{aligned}
||\Theta_{r,r'}\rangle\rangle \to ||\Theta_{r_1,r_2,r_1',r_2'}\rangle\rangle = \\
= \frac{1}{\sqrt{g_{r_1,r_2}}} \sum_{\substack{b_1,b_2 \in \{0,1\} \\ s_1,s_2}} ||r_1 + b_1, s_1; r_2 + b_2, s_2; r_2' + b_2, s_2; r_1' + b_1, s_1\rangle\rangle,
\end{aligned}
\tag{137}
$$

$$
\begin{aligned}
||\bar{\Theta}_{r,r'}\rangle\rangle \to ||\bar{\Theta}_{r_1,r_2,r_1',r_2'}\rangle\rangle = \\
= \frac{1}{\sqrt{g_{r_1,r_2}}} \sum_{\substack{b_1,b_2 \in \{0,1\} \\ s_1,s_2}} ||r_1 + b_1, s_1; r_1' + b_1, \bar{s}_1; r_2 + b_2, s_2; r_2' + b_2, \bar{s}_2\rangle\rangle,
\end{aligned}
\tag{138}
$$

$$
\begin{aligned}
||\tilde{\Theta}_{r,r'}\rangle\rangle \to ||\tilde{\Theta}_{r_1,r_2,r_1',r_2'}\rangle\rangle = \\
= \frac{1}{\sqrt{g_{r_1,r_2}}} \sum_{\substack{b_1,b_2 \in \{0,1\} \\ s_1,s_2}} ||r_1 + b_1, s_1; r_2 + b_2, s_2; r_1' + b_1, s_1; r_2' + b_2, s_2\rangle\rangle.
\end{aligned}
\tag{139}
$$

These obey relations equivalent to (108) and (109),

$$
\begin{aligned}
\left\langle\left\langle i_{r_1,r_2,r_1',r_2'} \,\middle\|\, i_{q_1,q_2,q_1',q_2'}\right\rangle\right\rangle = \frac{16}{g_{r_1,r_2}}\delta_{q_1 r_1}\delta_{q_2 r_2}\delta_{q_1' r_1'}\delta_{q_2' r_2'} \\
+ \frac{4}{\sqrt{g_{r_1 r_2}g_{q_1 q_2}}}\delta_{q_1,r_1+a}\delta_{q_2,r_2+b}\delta_{q_1',r_1'+a}\delta_{q_2',r_2'+b},
\end{aligned}
\tag{140}
$$

$$
\begin{aligned}
\left\langle\left\langle i_{r_1,r_2,r_1',r_2'} \,\middle\|\, j_{q_1,q_2,q_1',q_2'}\right\rangle\right\rangle = \frac{\delta_{r_1 r_2}}{6}\Big(2\delta_{q_1 r_1}\delta_{q_2 r_2}\delta_{q_1' r_1'}\delta_{q_2' r_2'} + \delta_{q_1,r_1+1}\delta_{q_2,r_2+1}\delta_{q_1',r_1'+1}\delta_{q_2',r_2'+1} \\
+ \delta_{q_1,r_1-1}\delta_{q_2,r_2-1}\delta_{q_1',r_1'-1}\delta_{q_2',r_2'-1}\Big),
\end{aligned}
\tag{141}
$$

with $a, b = \pm 1$, which can be used to obtain the interaction between neighboring pairs in successive layers

$$
v_{r_1+a,r_2+a,r_1'+a,r_2'+a} ||\Theta\bar{\Theta}_{r_1,r_2,r_1',r_2'}\rangle\rangle = \left(\frac{1}{4} - \frac{\delta_{r,r'}}{12}\right)||\Theta\bar{\Theta}_{r_1+a,r_2+a,r_1'+a,r_2'+a}\rangle\rangle,
\tag{142}
$$

$$
\begin{aligned}
v_{r_1+a,r_2-a,r_1'+a,r_2'-a} ||\Theta\bar{\Theta}_{r_1,r_2,r_1',r_2'}\rangle\rangle = \\
\left[\frac{1}{4} + \left(\frac{1}{2\sqrt{3}} - \frac{1}{4}\right)(\delta_{r,r'} + \delta_{r+a,r'-a})\right]||\Theta\bar{\Theta}_{r_1+a,r_2-a,r_1'+a,r_2'-a}\rangle\rangle.
\end{aligned}
\tag{143}
$$

Applying this to $||Q_0(0)\rangle\rangle$, given by (97), results in

$$||\overline{Q_0(t=1/2)}\rangle\rangle = \left(\sum_{R_1,R_2,R_1',R_2'=0}^{L/2-1} v_{2R_1-1,2R_2-1,2R_1'-1,2R_2'-1}\right)||Q_0(0)\rangle\rangle$$

$$= v_{-1,-1,-1,-1}||Q_0(0)\rangle\rangle = \frac{1}{\sqrt{3}}||\Theta\bar{\Theta}_{-1,-1,-1,-1}\rangle\rangle = \frac{1}{\sqrt{3}}||\Theta\bar{\Theta}_{-1,-1}\rangle\rangle,$$
(144)

i.e. the same initial condition we have for the NC-ST case. Since in (142) and (143) $r_1$ and $r_1'$ as well as $r_2$ and $r_2'$ are restricted to varying by the same amount, the subspace $\{||\Theta\bar{\Theta}_{r_1,r_2,r_1,r_2}\rangle\rangle\}$ is closed under time evolution. Thus, starting with (144) as the initial condition, the relation $||\Theta\bar{\Theta}_{r_1,r_2,r_1,r_2}\rangle\rangle = ||\Theta\bar{\Theta}_{r_1,r_2}\rangle\rangle$ effectively reduces the evolution relations (142) and (143) to the ones we had for NC-ST, (114) and (115). It follows that $\overline{C_2(r,t)}$ for the NC-T case is equal to the one obtained before for the NC-ST case. Equivalently to what happens for $\overline{C_1(r,t)}$, although the spatial homogeneity of the new circuit threatens the OTOC dynamics to be described by a four-dimensional process, initial conditions fix it to the two-dimensional process already known for NC-ST.

## 5.8 C-ST: averaged TOC, $\overline{C_1(r,t)}$

Lastly, we focus on the third instance of free fermion evolution, a restriction of the NC-ST case to a particle conserving process admitting spatio-temporal noise (C-ST). The difference from NC-ST lies in the new two-site unitaries conserving particle number, i.e. there are no anomalous terms $a_i a_j$ or $a_i^\dagger a_j^\dagger$ such that the $ph$ and $hp$ sectors of $u$ are null, where by $ph$ sector we mean those $|r,s\rangle\langle r,s|u|r',s'\rangle\langle r',s'|$ with $s=p$ and $s'=h$. The new unitaries are diagonal by blocks: $u_{r,r+1} = w_{r,r+1} \oplus w_{r,r+1}^*$, with $w$ $2\times 2$ random Haar-distributed unitary matrices 2.2.2. We start by computing $\overline{C_1(r,t)}$ and then $\overline{C_2(r,t)}$.

Knowing that, according to (69) in Section 5.2, $\overline{w_{\alpha\beta}^* w_{\mu\nu}} = \frac{1}{N}\delta_{\alpha\mu}\delta_{\beta\nu}$, the average of (79) becomes

$$\overline{\langle\alpha_1|u|\beta_1\rangle^*\langle\alpha_2|u|\beta_2\rangle} = \overline{\langle\alpha_1|w|\beta_1\rangle^*\langle\alpha_2|w|\beta_2\rangle}\delta_{s_{\alpha_1}s_{\beta_1}}\delta_{s_{\alpha_2}s_{\beta_2}} = \frac{1}{N}\delta_{\alpha_1\alpha_2}\delta_{\beta_1\beta_2}\delta_{s_{\alpha_1}s_{\beta_1}}\delta_{s_{\alpha_2}s_{\beta_2}},$$
(145)

where $N = \text{rank}(w) = 2$, with the difference from NC-ST being a restriction to the non-anomalous sectors $pp$ and $hh$. This results in the evolution operator (80) becoming

$$||\phi_r\rangle\rangle\langle\langle\phi_r|| \longrightarrow \Phi_r = \frac{1}{2}\sum_{\substack{x\in\{r,r+1\}\\x'\in\{r',r'+1\}}}\sum_s ||x,s;x,s\rangle\rangle\langle\langle x',s;x',s||.$$
(146)

The initial condition $||\overline{O_0^2(0)}\rangle\rangle = \sum_s ||0,s;0,s\rangle\rangle$ evolves under this as

$$||\overline{Q_0(1/2)}\rangle\rangle = \left(\sum_{R=0}^{L/2-1}\Phi_{2R-1}\right)||\overline{Q_0(0)}\rangle\rangle = \Phi_{-1}||\overline{Q_0(0)}\rangle\rangle = ||\phi_{-1}\rangle\rangle,$$
(147)

i.e. $||\overline{Q_0(1/2)}\rangle\rangle$ is equal to that of NC-ST and NC-T. Also, the interaction between neighboring pairs in successive layers reduces to

$$\Phi_{r\pm1}||\phi_r\rangle\rangle = \frac{1}{2}||\phi_{r\pm1}\rangle\rangle,$$
(148)

also analogous to $||\phi_r\rangle\rangle\langle\langle\phi_{r'}||\phi_{r\pm1}\rangle\rangle = 1/2$ valid for the NC-ST case. It follows that $\overline{C_1(r,t)}$ is the same we obtained for the previous two cases, with the restriction in the particle-hole sector in (146), when comparing to $||\phi_r\rangle\rangle\langle\langle\phi_r||$, being compensated by the different prefactor.

## 5.9   C-ST: averaged OTOC, $\overline{C_2(r,t)}$

Next, we obtain $\overline{C_2(r,t)}$ for the C-ST case, which differs from $\overline{C_2(r,t)}$ obtained for NC-T and C-ST.

**Time evolution of $||\overline{Q_0(t)}\rangle\rangle$**

As done for the NC-ST case, we start by computing $||\overline{Q_0(t)}\rangle\rangle$.

**Average over disorder realizations**   Particle conserving free fermion gates vary from their non-conserving counterparts. Being built as mentioned at the beginning of Section 5.8, their moments can be obtained directly from those of $2 \times 2$ unitary matrices $w$. Thus, their second moments $\overline{\langle \beta_1 |\, u\, |\alpha_1\rangle^* \langle \beta_2 |\, u\, |\alpha_2\rangle}$ are given by (145) while its fourth moment is

$$
\overline{\langle \beta_1 |\, u\, |\alpha_1\rangle^* \langle \beta_2 |\, u\, |\alpha_2\rangle^* \langle \beta_3 |\, u\, |\alpha_3\rangle \langle \beta_4 |\, u\, |\alpha_4\rangle}
$$
$$
= \overline{\langle \beta_1 |\, w\, |\alpha_1\rangle^* \langle \beta_2 |\, w\, |\alpha_2\rangle^* \langle \beta_3 |\, w\, |\alpha_3\rangle \langle \beta_4 |\, w\, |\alpha_4\rangle}\, \delta_{s_{\alpha_1} s_{\beta_1}} \delta_{s_{\alpha_2} s_{\beta_2}} \delta_{s_{\alpha_3} s_{\beta_3}} \delta_{s_{\alpha_4} s_{\beta_4}}, \quad (149)
$$

with the average of unitary matrices being (see Corollary 2.4 in [50])

$$
\overline{\langle \beta_1 |\, w\, |\alpha_1\rangle^* \langle \beta_2 |\, w\, |\alpha_2\rangle^* \langle \beta_3 |\, w\, |\alpha_3\rangle \langle \beta_4 |\, w\, |\alpha_4\rangle}
$$
$$
= \frac{1}{N(N^2-1)} \Big[ \delta_{\beta_1 \beta_4} \delta_{\beta_2 \beta_3} (N \delta_{\alpha_1 \alpha_4} \delta_{\alpha_2 \alpha_3} - \delta_{\alpha_1 \alpha_3} \delta_{\alpha_2 \alpha_4})
$$
$$
+ \delta_{\beta_1 \beta_3} \delta_{\beta_2 \beta_4} (N \delta_{\alpha_1 \alpha_3} \delta_{\alpha_2 \alpha_4} - \delta_{\alpha_1 \alpha_4} \delta_{\alpha_2 \alpha_3}) \Big]. \quad (150)
$$

with $N = \text{rank}(w) = 2$. With these changes, $v_{r,r'}$ given previously by (104) becomes

$$
v_{r,r'} \to v_{r,r'}^C = \frac{1 - \delta_{rr'}}{N^2} \Big[ ||\theta\rangle\rangle \langle\langle\theta|| \otimes (\Pi + \pi) + ||\tilde\theta\rangle\rangle \langle\langle\tilde\theta|| \otimes (\Pi + \tilde\pi) \Big]_{r,r'}
$$
$$
+ \frac{\delta_{rr'}}{N(N^2-1)} \Big[ N ||\theta\rangle\rangle \langle\langle\theta|| \otimes (\Pi + \pi)
$$
$$
- ||\theta\rangle\rangle \langle\langle\tilde\theta|| \otimes \Pi - ||\tilde\theta\rangle\rangle \langle\langle\theta|| \otimes \Pi + N ||\tilde\theta\rangle\rangle \langle\langle\tilde\theta|| \otimes (\Pi + \tilde\pi) \Big]_{r,r}, \quad (151)
$$

where we defined

$$
||\theta_{r,r'}\rangle\rangle = \sum_{x \in \{r,r+1\}} \sum_{x' \in \{r',r'+1\}} ||x, x', x', x\rangle\rangle, \quad (152)
$$

$$
||\tilde\theta_{r,r'}\rangle\rangle = \sum_{x \in \{r,r+1\}} \sum_{x' \in \{r',r'+1\}} ||x, x', x, x'\rangle\rangle, \quad (153)
$$

which respect

$$
\langle\langle i_{rr'} \,||i_{qq'}\rangle\rangle = 4\delta_{q,r} \delta_{q',r'} + \delta_{q,r\pm1} \delta_{q',r'\pm1}, \quad (154)
$$

$$
\langle\langle i_{rr'} \,||j_{qq'}\rangle\rangle = \delta_{r,r'} \Big( 2\delta_{q,r} \delta_{q',r'} + \delta_{q,r+1} \delta_{q',r'+1} + \delta_{q,r-1} \delta_{q',r'-1} \Big), \quad (155)
$$

with $i \neq j \in \{\theta, \bar\theta, \tilde\theta\}$. We also used

$$
\Pi = ||pppp\rangle\rangle \langle\langle pppp|| + ||hhhh\rangle\rangle \langle\langle hhhh|| \quad , \quad ||\Pi\rangle\rangle = ||pppp\rangle\rangle + ||hhhh\rangle\rangle \quad (156)
$$
$$
\pi = ||phhp\rangle\rangle \langle\langle phhp|| + ||hpph\rangle\rangle \langle\langle hpph|| \quad , \quad ||\pi\rangle\rangle = ||phhp\rangle\rangle + ||hpph\rangle\rangle \quad (157)
$$
$$
\tilde\pi = ||phph\rangle\rangle \langle\langle phph|| + ||hphp\rangle\rangle \langle\langle hphp|| \quad , \quad ||\tilde\pi\rangle\rangle = ||phph\rangle\rangle + ||hphp\rangle\rangle \quad (158)
$$

for which $PP' = \delta_{PP'}$ and $P||P\rangle\rangle = ||P\rangle\rangle$ holds, with $P$ and $P' \in \{\pi, \bar\pi, \tilde\pi\}$.

**Dynamics of $||\overline{Q_0(t)}\rangle\rangle$** Defining

$$||\Theta\tilde{\Theta}_{r,r'}\rangle\rangle = \Big( ||\theta_{r,r'}\rangle\rangle + ||\tilde{\theta}_{r,r'}\rangle\rangle \Big) \otimes ||\Pi\rangle\rangle \,, \tag{159}$$

we obtain from initial conditions

$$||\overline{Q_0(t=1/2)}\rangle\rangle = v^C_{-1,-1} ||Q_0(0)\rangle\rangle = \frac{1}{6} ||\Theta\tilde{\Theta}_{-1,-1}\rangle\rangle - \frac{1}{3} ||\tilde{\theta}_{-1,-1}\rangle\rangle \otimes ||\tilde{\pi}\rangle\rangle \,, \tag{160}$$

to which we can apply a second even layer of gates to obtain

$$||\overline{Q_0(t=1)}\rangle\rangle = \left( \sum_{R,R'=0}^{L/2-1} v^C_{2R,2R'} \right) ||Q_0(1/2)\rangle\rangle = (v^C_{-2,-2} + v^C_{-2,0} + v^C_{0,-2} + v^C_{0,0}) ||Q_0(1/2)\rangle\rangle$$

$$= \frac{1}{18}\Big( ||\Theta\tilde{\Theta}_{-2,-2}\rangle\rangle + ||\Theta\tilde{\Theta}_{0,0}\rangle\rangle \Big) + \frac{1}{24}\Big( ||\Theta\tilde{\Theta}_{-2,0}\rangle\rangle + ||\Theta\tilde{\Theta}_{0,-2}\rangle\rangle \Big)$$

$$- \frac{1}{9}\Big( ||\tilde{\theta}_{-2,-2}\rangle\rangle + ||\tilde{\theta}_{0,0}\rangle\rangle \Big) \otimes ||\tilde{\pi}\rangle\rangle - \frac{1}{12}\Big( ||\tilde{\theta}_{-2,0}\rangle\rangle + ||\tilde{\theta}_{0,-2}\rangle\rangle \Big) \otimes ||\tilde{\pi}\rangle\rangle \,. \tag{161}$$

One can check, making use of (154) and (155), that $||\Theta\tilde{\Theta}_{r,r'}\rangle\rangle$ and $||\tilde{\theta}_{r,r'}\rangle\rangle \otimes ||\tilde{\pi}\rangle\rangle$ evolve under $v^C_{r,r'}$ as

$$v^C_{r+a,r'+a} ||\Theta\tilde{\Theta}_{r,r'}\rangle\rangle = \left( \frac{1}{4} + \frac{\delta_{r,r'}}{12} \right) ||\Theta\tilde{\Theta}_{r+a,r'+a}\rangle\rangle \,, \tag{162}$$

$$v^C_{r+a,r'-a} ||\Theta\tilde{\Theta}_{r,r'}\rangle\rangle = \left( \frac{1}{4} - \frac{\delta_{r,r'}}{12} \right) ||\Theta\tilde{\Theta}_{r+a,r'-a}\rangle\rangle \,, \tag{163}$$

$$v^C_{r\pm a,r'\pm a} ||\tilde{\theta}_{r,r'}\rangle\rangle \otimes ||\tilde{\pi}\rangle\rangle = \left( \frac{1}{4} + \frac{\delta_{r,r'}}{12} \right) ||\tilde{\theta}_{r\pm a,r'\pm a}\rangle\rangle \otimes ||\tilde{\pi}\rangle\rangle \,, \tag{164}$$

with $a = \pm 1$. These relations control the mixing occurring between neighboring pairs in successive time steps and imply that both $\{||\Theta\tilde{\Theta}_{r,r'}\rangle\rangle\}$ and $\{||\tilde{\theta}_{r,r'}\rangle\rangle \otimes ||\tilde{\pi}\rangle\rangle\}$ are closed under time evolution, besides being orthogonal to one another, since $\langle\langle\Pi ||\tilde{\pi}\rangle\rangle = 0$. Thus, we can write

$$||\overline{Q_0(t)}\rangle\rangle = \sum_{R=0}^{L/2-1} \left( \frac{||\Theta\tilde{\Theta}_{2R,2R'}\rangle\rangle \langle\langle\Theta\tilde{\Theta}_{2R,2R'}||}{8(2+\delta_{2R,2R'})} + \frac{||\tilde{\theta}_{2R,2R'}\rangle\rangle \langle\langle\tilde{\theta}_{2R,2R'}|| \otimes ||\tilde{\pi}\rangle\rangle \langle\langle\tilde{\pi}||}{8} \right) ||\overline{Q_0(t)}\rangle\rangle \,, \tag{165}$$

valid for $t$ integer, becoming valid for $t$ half-integer under $2R \to 2R-1$ and $2R' \to 2R'-1$. The prefactors originate from $\langle\langle\Theta\tilde{\Theta}_{r,r'} ||\Theta\tilde{\Theta}_{q,q'}\rangle\rangle = 8(2 + \delta_{r,r'})\delta_{r,q}\delta_{r',q'}$ and $\left\langle\left\langle\tilde{\theta}_{r,r'} ||\tilde{\theta}_{q,q'}\right\rangle\right\rangle \langle\langle\tilde{\pi} ||\tilde{\pi}\rangle\rangle = 8\delta_{r,q}\delta_{r',q'}$, where the indices are restricted to having the same parity. Ahead, we will see that only states of the type $||\Theta\tilde{\Theta}_{r,r'}\rangle\rangle$ contribute to the OTOC and thus we can ignore $\{||\tilde{\theta}_{r,r'}\rangle\rangle \otimes ||\tilde{\pi}\rangle\rangle\}$. Although $||\Theta\tilde{\Theta}_{r,r'}\rangle\rangle$ are analogous to the states $||\Theta\bar{\Theta}_{r,r'}\rangle\rangle$ appearing in the NC-ST case, the two are inherently different: while $||\Theta\tilde{\Theta}_{r,r'}\rangle\rangle$ includes, besides $||\alpha\beta\beta\alpha\rangle\rangle$, states of the type $||\alpha\beta\alpha\beta\rangle\rangle$ which do not appear in NC-ST due to PH symmetry, it leaves out states of the kind $||\alpha\bar{\alpha}\beta\bar{\beta}\rangle\rangle$, which do not appear when averaging over the gates.

Having $\{1/[8(2+\delta_{r,r'}) ||\Theta\tilde{\Theta}_{r,r'}\rangle\rangle]\}$ as the natural basis to use, we use (162) and (163) to obtain, for $t \geq 1/2$, an expression equivalent to (117) with the changes $\langle\langle\Theta\bar{\Theta}_{r,r'}|| \to \langle\langle\Theta\tilde{\Theta}_{r,r'}||$ and $m_{r,r'} \to m^C_{r,r'}$, with $m^C_{r,r'}$ coefficient matrices given by

$$m^C_{r,r} = \begin{pmatrix} \frac{1}{3} & \frac{1}{4} \\ \frac{1}{4} & \frac{1}{3} \end{pmatrix} \quad , \quad m^C_{r,r+2} = (m^C_{r,r-2})^T = \begin{pmatrix} \frac{1}{4} & \frac{1}{4} \\ \frac{1}{6} & \frac{1}{4} \end{pmatrix} \quad , \quad m^C_{r,r'} = \frac{1}{4}\begin{pmatrix} 1 & 1 \\ 1 & 1 \end{pmatrix}, \tag{166}$$

with the last $m^C_{r,r'}$ being valid for $|r - r'| > 2$. Applying this twice gives an expression equivalent to (119) with $\langle\langle \Theta\bar{\Theta}_{r,r'}|| \rightarrow \langle\langle \Theta\tilde{\Theta}_{r,r'}||$ and $M_{r,r'} \rightarrow M^C_{r,r'}$, with the new coefficient matrices $M^C_{r,r'}$ being

$$M^C_{r,r} = \frac{1}{144} \begin{pmatrix} 16 & 21 & 9 \\ 21 & 44 & 21 \\ 9 & 21 & 16 \end{pmatrix} \quad , \quad M^C_{r,r+2} = (M^C_{r,r-2})^T = \frac{1}{32} \begin{pmatrix} 1 & 2 & 1 \\ \frac{14}{9} & \frac{11}{3} & 2 \\ \frac{2}{3} & \frac{14}{9} & 1 \end{pmatrix}$$

$$M^C_{r,r+4} = (M^C_{r,r-4})^T = \frac{1}{16} \begin{pmatrix} 1 & 2 & 1 \\ 2 & 4 & 2 \\ \frac{2}{3} & 2 & 1 \end{pmatrix} \quad , \quad M^C_{r,r'} = \frac{1}{16} \begin{pmatrix} 1 & 2 & 1 \\ 2 & 4 & 2 \\ 1 & 2 & 1 \end{pmatrix}, \tag{167}$$

where the last $M^C_{r,r'}$ is valid for $|r - r'| > 4$. At last, starting with (161) as the initial condition, $||\overline{Q_0(t)}\rangle\rangle$ is given by an analogue of the recursive expression (119), with $\langle\langle \Theta\bar{\Theta}_{r,r'}|| \rightarrow \langle\langle \Theta\tilde{\Theta}_{r,r'}||$ and $M_{r,r'} \rightarrow M^C_{r,r'}$, alongside (165). Although the details surrounding C-ST and NC-ST are different, the overall structure appearing in the dynamics of $||\overline{Q_0(t)}\rangle\rangle$ is similar. Namely, the dependence on the nearest and second nearest neighbors is the same almost everywhere, i.e. $M^C_{r,r'} = M_{r,r'}$ for $|r - r'| > 4$.

**Exact result for $\overline{C_2(r,t)}$**

Now that we know the dynamics of $||\overline{Q_0(t)}\rangle\rangle$ we can obtain that of the OTOC.

For $t = 0$ (121) holds. For $t \geq 1$ we can decompose $\overline{C_2(r,t)}$ using (165) to obtain

$$\overline{C_2(r,t)} = \langle\langle Q_r|| \, S \, ||\overline{Q_0(t)}\rangle\rangle$$
$$= \langle\langle Q_r|| \, S \left( \sum_{R,R'=0}^{L/2-1} \frac{1}{8(2 + \delta_{r,r'})} ||\Theta\tilde{\Theta}_{2R,2R'}\rangle\rangle \left\langle\langle \Theta\tilde{\Theta}_{2R,2R'} \, \Big\| \overline{Q_0(t)}\right\rangle\right\rangle \right)$$
$$= \frac{1}{6} \sum_{R=0}^{L/2-1} (\delta_{r,2R} + \delta_{r,2R+1}) \left\langle\langle \Theta\tilde{\Theta}_{2R,2R} \, \Big\| \overline{O_0^2(t)}\right\rangle\right\rangle. \tag{168}$$

Note that, as we already suggested, the term $||\tilde{\theta}_{r,r'}\rangle\rangle \otimes ||\tilde{\pi}\rangle\rangle$ coming from (165) does not contribute to the OTOC, since the particle-hole sector of $S||Q_r\rangle\rangle$ is $||\Pi + \pi\rangle\rangle$ and $\langle\langle \Pi + \pi \, ||\tilde{\pi}\rangle\rangle = 0$.

Defining

$$K^C_{r,r'}(t) = \frac{1}{6} \left\langle\langle \Theta\tilde{\Theta}_{r,r'} \, \Big\| \overline{Q_0(t)}\right\rangle\right\rangle, \tag{169}$$

analogous to (123), the OTOC is then $\overline{C_2(2R,t)} = \overline{C_2(2R+1,t)} = K^C_{2R,2R}(t)$. Finally, we can use (161) to get the OTOC at $t = 1$,

$$\overline{C_2(2R, t=1)} = \overline{C_2(2R+1, t=1)} = \frac{2}{9}\Big(\delta_{R,-1} + \delta_{R,0}\Big), \tag{170}$$

and, at subsequent time steps, the dynamics of $\overline{C_2(r,t)}$ is determined by that of $\left\langle\langle \Theta\bar{\Theta}_{r,r'} \, \Big\| \overline{O_0^2(t)}\right\rangle\right\rangle \Leftrightarrow K^C_{r,r'}(t)$, we just saw to be given by (119) with the changes $\langle\langle \Theta\bar{\Theta}|| \rightarrow \langle\langle \Theta\tilde{\Theta}||$ and $M \rightarrow M^C$. This is, for $t > 1$ the OTOC is determined by (125) with the changes $K_{r,r'}(t) \rightarrow K^C_{r,r'}(t)$ and $M_{r,r'} \rightarrow M^C_{r,r'}$, given by (159) and (167), respectively.

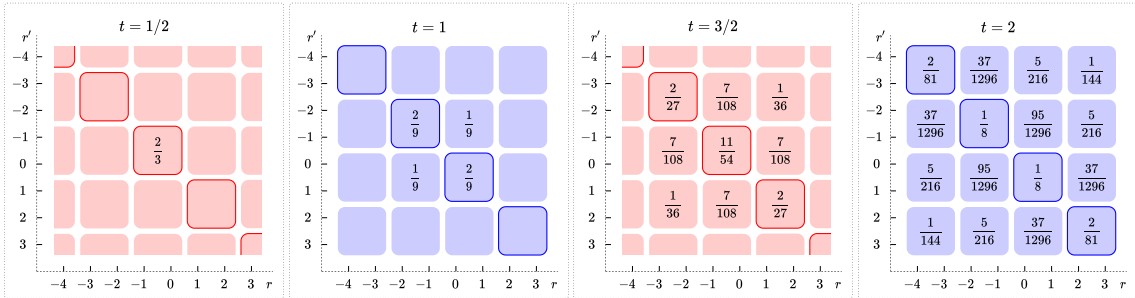

Figure 9: The values in the boxes correspond to $K_{r,r'}^C(t)$ in the $r - r'$ plane, shown here for $t \in \{1/2, 1, 3/2, 2\}$ (empty boxes have zero value). Starting with $K_{r,r'}^C(t) = 2\delta_{r,r',0}$ as the initial condition, we evolve this by applying (117) successively (with $\langle\langle\Theta\bar{\Theta}|| \to \langle\langle\Theta\tilde{\Theta}||$ and $M \to M^C$, given by (159) and (167), respectively). The highlighted diagonal gives $\overline{C_2(r,t)}$ for the C-ST case. This picture is equivalent to Fig. 8, valid for NC-ST and NC-T.

**Continuum limit of $\overline{C_2(r,t)}$** Since the bulk evolution of $K_{r,r'}^C(t)$ is equal to that of $K_{r,r'}(t)$, i.e. $m_{r,r'}^C|_{|r-r'|>2} = m_{r,r'}|_{|r-r'|>2}$, the continuum limit of the OTOC for the C-ST case is analogous to the one obtained for NC-ST. The differences of the diagonal with regards to the NC-ST case translate to

$$c \to c^C = 3/2, \tag{171}$$

$$m'_{r,r} \to m'^C_{r,r} = \frac{1}{6}\begin{pmatrix} 2 & 1 \\ 1 & 2 \end{pmatrix}, \tag{172}$$

$$f \to f^C = 2. \tag{173}$$

In this case, the anisotropy of $m'^C_{r,r}$ leads to diffusion within the diagonal with weighting factor $f^C = 2$.

This results in the OTOC being given by (132), with the changes $c = \sqrt{3}/2 \to c^C = 3/2$ and $f = 1/2 \to f^C = 2$. Thus, in the continuum limit, the OTOC is a Gaussian with broadening width $\sigma(t) = \sqrt{t}$, equal to the NC-ST and NC-T cases apart from the normalization which is $A(t) \to A^C(t) = f^C/f A(t) = f^C A/(2\sqrt{2\pi t})$.

## 5.10 Comparing NC-ST, NC-T and C-ST

The NC-T case differs from NC-ST by the way the circuit is built − instead of having spatial disorder, a gate is randomly chosen at each half time step and applied to all pairs of sites (Fig. 2). In principle, this could lead the dynamics of $\overline{C_1}$ and $\overline{C_2}$ to arise from respectively 2D and 4D processes (Sections 5.6 and 5.7), but the initial condition reduces them to the 1D and 2D processes already known for NC-ST.

To obtain the particle conserving dynamics (C-ST), the free fermion gates used in the NC-ST case, built from orthogonal matrices, are replaced by gates built out of unitary matrices. Since the average of two orthogonal and two unitary gates needed to obtain $\overline{C_1}$ coincide, the TOC is equal to the one obtained for the NC-ST case (Section 5.8). The picture is different for $\overline{C_2}$ − it is the only correlator which differs from the NC-ST results out of the cases studied. This happens because the average of four gates needed to compute $\overline{C_2}$ is different for orthogonal and unitary matrices (Section 5.9). However, this does not change the fundamental diffusive nature of the process. It merely leads to a different normalization in the continuum limit.

## 5.11 Continuum limit of $\overline{\mathcal{C}(r,t)}$

Finally, we can take into account the contributions of $\overline{C_1(r,t)}$ and $\overline{C_2(r,t)}$ in the continuum limit, given respectively by (93) and (132), to obtain

$$\overline{\mathcal{C}(2R,t)} = \overline{\mathcal{C}(2R+1,t)} = \frac{1}{2^4}\left[\frac{A}{\sqrt{4\pi t}}\exp\left(-\frac{(2R)^2}{4t}\right) - f\frac{A}{4\pi t}\exp\left(-\frac{(2R)^2}{2t}\right)\right], \quad (174)$$

where $A = 2$ and $f = 1/2$ for the NC-ST and NC-T cases and $f \to f^C = 2$ for the C-ST case.

Although $\overline{\mathcal{C}(r,t)}$ is typically given by the sum of two Gaussians with different standard deviation, in the infinite time limit it reduces to the first one, i.e. $\lim_{t\to\infty}\overline{\mathcal{C}(r,t)} = \overline{C_1(r,t)}/2^3$ since $\lim_{t\to\infty}\overline{C_2(r,t)}/\overline{C_1(r,t)} = 0$. This occurs because the normalization of $\overline{C_1(r,t)}$ is constant while that of $\overline{C_2(r,t)}$ is $\propto 1/\sqrt{t}$.

## 5.12 Numerical results

In the previous sections, we obtained analytical expressions for the TOC and OTOC for three instances of free fermion evolution: NC-ST, C-ST and NC-T. In the following, we compare these results with the respective approximate expressions in the continuum limit and also with data obtained from simulations. We also realize a fourth instance of free fermion evolution numerically − a temporally homogeneous case where randomness appears only along the space direction (NC-S). While the analytical results take into account all possible evolutions of the system, i.e. all possible circuits, the simulations approximate this by an average over $N_r = 4000$ disorder realizations. We look at a system with $L = 100$ using $C_1(r,0) = C_2(r,0) = 2\delta_{r,0}$ as the initial condition.

We begin with the analytical results, plotted in Fig. 10. We see that the profile of $\overline{C_1(r,t)}$ and $\overline{C_2(r,t)}$ at fixed time steps is given by a Gaussian whose width broadens with time according to $\sigma(t) = \sqrt{2t}$ and $\sigma(t) = \sqrt{t}$, respectively. We see that $\overline{C_1}$ dominates over $\overline{C_2}$. To compare these results with the data obtained from simulations we draw the profile of both $\overline{C_1(r,t)}$ and $\overline{C_2(r,t)}$ at fixed time steps.

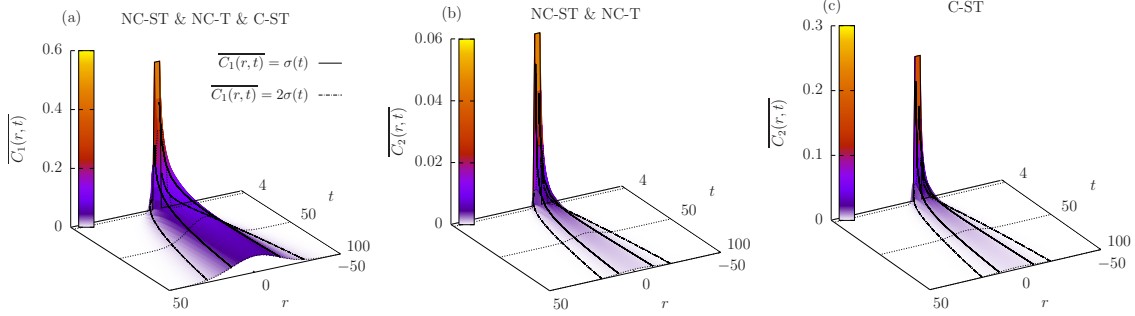

Figure 10: (a) Height and colour maps of the exact $\overline{C_1(r,t)}$ in the $r-t$ plane for the NC-ST, NC-T and C-ST cases, given by (91). (b) Exact $\overline{C_2(r,t)}$ for the NC-ST and NC-T cases, given by (124) for $t = 1$ and (125) for $t > 1$. (c) Exact $\overline{C_2(r,t)}$ for the C-ST case, given by (170) for $t = 1$ and by an expression equivalent to (125) for $t > 1$, with the changes $K \to K^C$ and $M \to M^C$ given respectively by (169) and (167). These results are for a system with $L = 100$ and periodic boundary conditions. The results are shown for $1 \le t \le 100$. The black curves indicate $\sigma(t)$ and $2\sigma(t)$, with $\sigma(t) = \sqrt{2t}$ for $\overline{C_1}$ and $\sigma(t) = \sqrt{t}$ for $\overline{C_2}$.

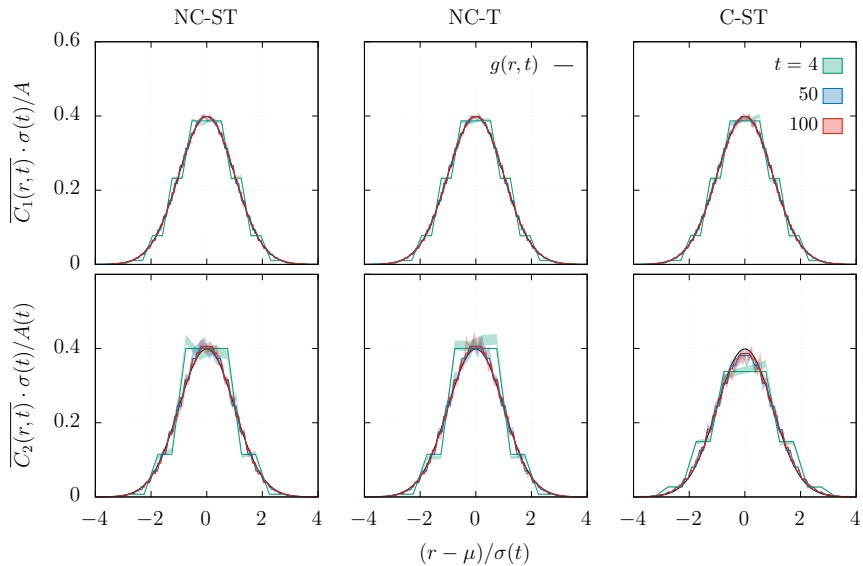

Figure 11: Rescaled TOC, $\overline{C_1(r,t)} \cdot \sigma(t)/A$, (top) and OTOC, $\overline{C_2(r,t)} \cdot \sigma(t)/A(t)$, (bottom) as a function of $(r-\mu)/\sigma(t)$ at three fixed time steps $t \in \{4, 50, 100\}$ for a system with $L = 100$ and for the three cases NC-ST (left), NC-T (centre) and C-ST (right). Full lines correspond to the exact calculations seen in Fig. 10 and the colored regions to data obtained from simulations averaging over $N_r = 4000$ disorder realizations. Also, $g(r,t) = 1/(\sqrt{2\pi})\exp\left(-(r-\mu)^2/2\sigma^2(t)\right)$. For each case, the appropriate standard deviation $\sigma(t)$ and normalization $A$ or $A(t)$ are used. For $\overline{C_1(r,t)}$, $A = 2$ and $\sigma(t) = \sqrt{2t}$ for all cases. For $\overline{C_2(r,t)}$, $\sigma(t) = \sqrt{t}$ for all cases, $A(t) = 1/(2\sqrt{2\pi t})$ for NC-ST and NC-T and $A(t) = 2/(\sqrt{2\pi t})$ for C-ST. The deviation $\mu = 1/2$ appears because the brickwall structure of the circuit leads the TOC and OTOC to be centred around $\mu$ instead of 0.

Fig. 11 is a comparison of the analytic results and numerical simulation data for the three different cases NC-ST, NC-T and C-ST. The continuum limit in each case is tested by checking whether $\overline{C_1}$ and $\overline{C_2}$ respect (93) and (132), respectively. This translates to $\overline{C_1(r,t)}\,\sigma(t)/A$ and $\overline{C_2(r,t)}\,\sigma(t)/A(t)$ collapsing to the Gaussian $g(r,t) = \frac{1}{\sqrt{2\pi}}\exp\left(-\frac{(r-\mu)^2}{2\sigma^2(t)}\right)$, with $A$ or $A(t)$ and $\sigma(t)$ given for each case in the caption of Fig. 11. We see in Fig. 11 that, for small time steps ($t = 4$), the discrete data does not follow a Gaussian, as expected, while for higher time steps it does. Indeed, for a reasonable system size ($L = 100$), the continuum limit quickly becomes a good approximation for either the TOC or the OTOC. For time steps significantly larger than $t = 100$, periodic boundary conditions cause the discrete data to diverge from the continuum limit expectation.

We have established that the analytical SB results, the data from simulations and the continuum limit expressions are in good mutual agreement. Before proceeding to analyse the numerical results obtained for the NC-S case, we provide some indication of the fluctuations present in both $C_1(r,t)$ and $C_2(r,t)$ by showing a single realization for each case of free fermion evolution in Fig. 12.

### 5.12.1 NC-S: randomness in space alone

We now present numerical results for NC-S for $L = 100$ and $N_r = 4000$. The circuit for NC-S is built as shown in Fig. 2 (c): we construct two random layers, one even and one odd, and apply them repeatedly, such that there is discrete time translation symmetry.

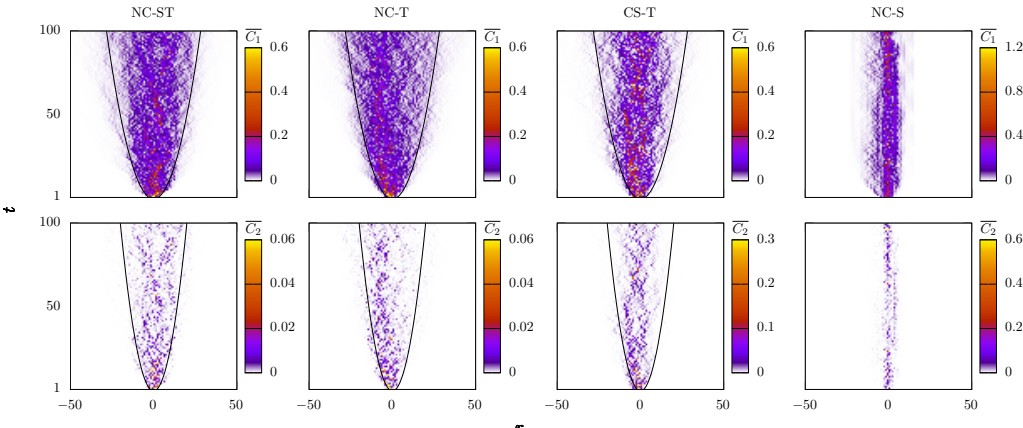

Figure 12: Single disorder realization of $C_1(r,t)$ (top) and $C_2(r,t)$ (bottom) as a colour map in the $r - t$ plane for NC-ST (left), NC-T (centre left), CS-T (centre right) and NC-S (right). The black curves are $r - \mu = 2\sigma(t)$, i.e. $t = (r - \mu)^2/8$ for $\overline{C_1}$ and $t = (r - \mu)^2/4$ for $\overline{C_2}$. These results are for a system with $L = 100$ and periodic boundary conditions. The deviation $\mu$ appears as it did in Fig. 11.

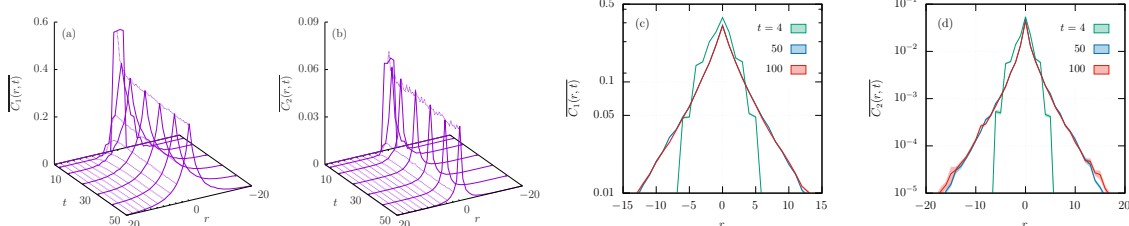

Figure 13: (a) and (b) Height map of $\overline{C_1(r,t)}$ and $\overline{C_2(r,t)}$, respectively, in the $r - t$ plane for $1 \leq t \leq 50$, in the NC-S case. (c) and (d) Profile of $\overline{C_1(r,t)}$ and $\overline{C_2(r,t)}$ on a logarithmic scale, obtained by fixing $t$ to 4, 50 and 100 in panels (a) and (b). The data results from an average over $N_r = 4000$ simulations, for a system with $L = 100$ and periodic boundary conditions.

We see in Fig. 13 that for the NC-S case both the TOC and OTOC are localized in space, decaying exponentially around $r = 0$ − see Fig. 13(b,d). This phenomenon, arising from the disordered landscape, is a discrete time analogue of Anderson localization. This picture is unlike the diffusive behaviour seen for quadratic fermion evolution and the usual ballistic behaviour. Furthermore, this hints that the diffusive behaviour observed for NC-ST, NC-T and also C-ST is due to randomness in time alone.

## 6 Deviations from a 1D random walk

Here, we compare the results obtained Sections 4 and 5.

In the single-body basis (Section 5) we saw that $\overline{\mathcal{C}(r,t)} = 2[\overline{C_1(r,t)} - \overline{C_2(r,t)}]$ (62), with $\overline{C_1}$ and $\overline{C_2}$ correlators of single-body observables. For details on the calculations of $\overline{C_1}$ and $\overline{C_2}$ see Sections 5.4 and 5.5, respectively. While $\overline{C_1(r,t)}$ (91) is exactly the probability for a random walker to reach position $r$ after time $t$, $\overline{C_2(r,t)}$ accounts for the deviations of $\overline{\mathcal{C}(r,t)}$ from a 1D random walk. Specifically, $\overline{C_2}$ is the diagonal of a 2D process close

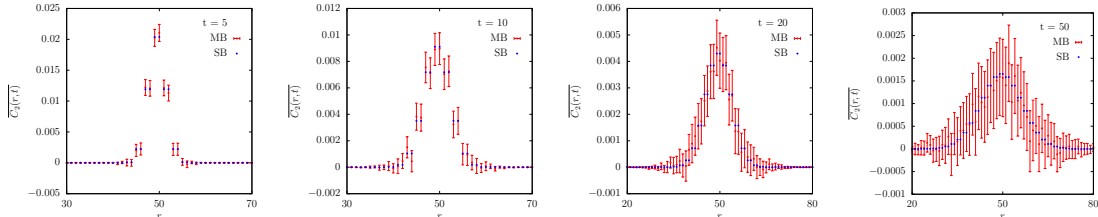

Figure 14: In blue, $\overline{C_2(r,t)}$ and, in red, $\overline{C_1(r,t)} - \overline{\mathcal{C}(r,t)}/2$ for a system with $L = 100$ and time fixed at $t = 5, 10, 20$ and $50$. Here, $\overline{C_1(r,t)}$ and $\overline{C_2(r,t)}$ are given analytically by (91) and (125), respectively, and $\overline{\mathcal{C}(r,t)}$ is obtained as described in Fig. 5(a), but using instead $10^6$ string evolution histories.

to a 2D random walk, given by (125). A 2D random walk can be decomposed into two independent 1D random walks. In the 2D random process behind $\overline{C_2(r,t)}$, the deviations from a 2D random walk occur precisely when the walker in 2D is in the main diagonal or the diagonals immediately above and bellow the main one, which corresponds precisely to the two 1D walkers meeting or occupying adjacent sites. As discussed before, in the many-body calculation (Section 4), these are precisely the points where the two walkers deviate from being independent. Let us emphasize that $\overline{C_2(r,t)}/\overline{C_1(r,t)} = 1/\sqrt{t} \to 0$ when $t \to \infty$, i.e. the corrections are in fact a subleading term and tend to become less significant in time. This is expected since, as the string grows, its endpoints/walkers are less likely to meet.

The results obtained for $\overline{\mathcal{C}(r,t)}$ in the single and many-body basis match. This is shown in Fig. 14, which compares the SB $\overline{C_2}$ given by (125) and the deviations of the many-body $\overline{\mathcal{C}(r,t)}$ from a 1D random walk, i.e. $\overline{C_1(r,t)} - \overline{\mathcal{C}(r,t)}/2$, where $\overline{C_1(r,t)}$ is exactly the 1D random walk given by (91) and $\overline{\mathcal{C}(r,t)}$ the many-body result obtained as described in Fig. 5. The agreement observed indicates, as suspected, that $\overline{C_2(r,t)}$ accounts for the interaction and annihilation of random walkers.

# 7 Discussion and Outlook

Matchgate and free-fermion circuits are of wide interest, e.g. in the context of computational complexity [51–59], for investigating effects of measurement [60–63] or other non-unitary processes [64–66]. In this work, we have used this class of circuits to provide a derivation of (and intuition for) a peculiar phenomenon that is apparently generic in several types of free-fermion systems and in systems that can be mapped to free fermions.

The diffusive spreading of information and accompanying $S(t) \sim \sqrt{t}$ behavior for free fermions subject to temporal noise appears to be very robust − it has been observed both for circuits [34] and for Hamiltonian evolution [2, 32, 33, 35, 36], and both for continuous-time noise [32, 33, 36] and for discrete-time randomness [34, 35]. Thus, a generic derivation of the phenomenon is highly desirable. The present work provides such a derivation. It also provides physical intuition, grounded in the Pauli string calculation, for the fact that free fermions scramble information poorly. While this setting admittedly only treats the discrete-time case, it is widely applicable because all inessential features are stripped out of the circuit model, laying bare the essence of a highly nontrivial but robust phenomenon.

Our expectation is that the results obtained for the circuit model of free fermions generalize to the case of noisy free fermion Hamiltonian evolution. With this goal in mind, it may be interesting to revisit the stochastic free-fermion model of Refs. [49, 67, 68].

# Acknowledgements

BD acknowledges support by FCT through Grant No. UIDB/04540/2020. BD and PR acknowledge support by FCT through Grant No. UID/CTM/04540/2019.

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
