# Peer review of "Diffusive Operator Spreading for Random Unitary Free Fermion Circuits"

_SciPost Physics_

## Round 1 · Referee Report · Anonymous (Referee 1) · 2022-8-29

Strengths

1- very detailed analytical presentation (no questions left unanswered) 2- nice summary of the properties of Gaussian circuits in general 3- thorough analysis of the OTOC dynamics in four different types of Gaussian circuits

Weaknesses

1- the periodic switching between the Pauli-operator language (what the authors refer to as 'many-body calculation') and the matrix representation of Gaussian circuits (fermions) makes the paper hard to read 2- the object of analysis (OTOCs) is rather specific/narrow and the main finding of the authors (diffusive spreading) is, in my opinion, not overwhelming 3- The result of diffusive spreading is somewhat expected from my point of view (see below for details) and the crucial point here is the absence of energy conservation in a circuit (which the authors mention explicitly). Thus I can't follow their logic when comparing the random circuit to a translational invariant system or an Anderson localized system.
4- The way the analytical calculations are presented makes it really hard to follow the manuscript. At several points in the later sections, I was asking myself 'why I should actually read and try to understand the next 3 pages of equations'.

Report

Overall, the authors provide a thorough analysis of the OTOC in Gaussian random fermion circuits. As far as I can judge, the calculations are correct and the obtained results match with the intuition (at least of this particular referee). Having that said, the paper deserves publication in a scientific journal but from my point of view not in SciPost physics. The results seem (i) too specific to attract a broader scientific audience, and (ii) not groundbreaking or providing a breakthrough in order to advance research on quantum dynamics with quantum circuits. Therefore I suggest publication in SciPost Physics Core, which is better suited to the specialised and technical nature of the work.

Apart from being quite technical and specific, the main reason for the above decision is the rather unsurprising result of observing a diffusive spreading of information for random Gaussian fermion circuits. The authors mention in the beginning of their introduction that random circuits represent a good proxy for certain types of quantum dynamics, for which energy conservation is not required. A few lines down in the text they then seem surprised that random Gaussian circuits act quite differently from two paradigmatic Hamiltonian systems (ballistic spreading and Anderson localization), which are known to require energy conservation at the essence!
In such cases, one should expect a diffusive dynamics! The simplest example being a quadratic fermion Hamiltonian (disordered or not) subject to a random fluctuating onsite potential, equivalent to local dephasing (e.g. discussed in the last Section of Ref.[33] or Phys. Rev. Research 4, 013109 (2022) or Phys. Rev. B 103, 184202 (2022)). Diffusive spreading of information is the generically expected behavior for systems that are updated (i) by local operators and (ii) do not obey any particular symmetries.

---

## Round 1 · Referee Report · Ivan Khaymovich (Referee 2) · 2022-9-4

Strengths

1 - interesting and relevant problem in focus
2 - developed analytical approach to solve it and its prominent application to further relevant time-dependent problems
3 - unexpected result of a diffusion in non-interacting (free-fermion) random circuits

Weaknesses

1 - usage of several notations for the same objects interchangeably
2 - missing definitions of some objects
3 - misprints

Report

The authors of this manuscript consider a free fermionic model, evolved under quantum random circuits that mimic the form of the unitary operator, being quadratic (non-interacting) in the creation/annihilation operators. Comparing to the usual Haar-measure distributed random quantum circuit (RQC) results, the authors find a striking and unexpected difference in the propagation of correlations. While in Haar-measure RQC the correlations propagate as a light cone with diffusively spreading front (i.e. as a drift-diffusion process), in the free-fermion case the drift is absent and the correlations propagate only via diffusion. Considering an exemplary initial local $\sigma_z$-operator, the authors uncover the origin of the above striking difference: indeed, the propagation of the $\sigma_z$-operator in a free-fermion RQC is peculiar: it contains $\sigma_x$- and $\sigma_y$-operators only at the front, separating $\sigma_z$-bulk from $I$-surroundings. It means that the scrambling of the degrees of freedom in such a free-fermionic model appears only locally and is quite similar to some string-like confinement of mesons in QED and kinetically constrained models.

The central result of the paper is an exact analytical calculation of the many-body out-of-time-ordered correlator (OTOC) in a free-fermion RQC model. With help of this calculation the authors derive the diffusive growth of entanglement and explain the difference between free-fermion RQC and their generic Haar-measure unitary counterparts.

I have several questions and comments to the authors: 1) My main question is related to the form of the operator spreading, shown in Figs. 1(c) and 4(a,b). Indeed, as discussed in Sec. 4.2.3 the initially local operator can be expanded at any times into the spin-strings of the only form of $\ldots I\otimes \sigma_i \otimes \sigma_z \ldots \sigma_z \otimes \sigma_j \otimes I \ldots$ As far as I see from the discussion (especially in Secs. 4.2.2 and 4.2.3), it is true only for the initial operator of the form $\ldots I\otimes \sigma_z \otimes I \ldots$ What will happen for other types of local operators? Straightforwardly one can understand what will happen with the initial operator, given by a product of only I and $\sigma_z$ single-spin operators, because it is a superposition of the previous results, but what is more interesting is how the operators like $\ldots I\otimes \sigma_x \otimes I \ldots$ will spread.

2) The follow-up of the previous question is the following: it seems for me that the $\sigma_z$- and $I$-operators may be considered as fermionic particles and holes according to their behavior, while the $\sigma_{x,y}$-operators behave similarly to Majorana modes (two Majorana form one fermion) or like half-quantum vortices as their movement is accompanied by forming a string of z-particles (similarly to the confinement of mesons in QED). If this analogy is correct, then one can understand all the (unphysical) complexities, appearing from the initial operator $\ldots I\otimes \sigma_x \otimes I \ldots$, as we start from an unpaired-Majorana state. Please clarify whether this analogy is correct and, if yes, please mention this in the manuscript.

3) The authors consider several cases of the spatio-temporal disorder with or without particle number conservation. What about the case of the temporal disorder with the particle number conservation? Is it different from the 3 considered cases?

4) The authors stick to the superconducting (particle-hole symmetric) case in Eq. (2) and further. What will happen if this particle-hole symmetry (together with the parity conservation) is broken? Do you expect to see anyonic spreading instead of Majorana mode one?

5) In general from the presentation point of view, I suggest the authors to rearrange the manuscript in such a way to avoid repetitive explanation of the same concepts, such as Separation in parts in Eq. (17-19) and (33-36), Pauli strings, formula for OTOC via $a_S(t)$ and many others, mentioned in Sec. 3 and then properly defined in Secs. 4-6.

6) Can one understand the result for free-fermion RQC as the Haar-measure one with zero drift velocity (see Fig. 6)? If yes, please clarify the origin of this.

The other comments are minor and related to the presentation of the work, therefore I have put them to the "Requested changes".

Requested changes

  1. Please, first, consider all the questions and suggestions of the report properly.
  2. In page 3 after Fig. 1 it might be useful to mention a drift-diffusion process instead of "a ballistic front that itself spreads diffusively".
  3. Please put your manuscript in the context of Refs. [26-29] on free-fermion Hamiltonian models: what has been done there?
  4. Before Eq. (4) please define $\langle \ldots \rangle$ via the density matrix and the trace in order to clarify the derivation of r.h.s. of Eq. (4).
  5. Just before Eq. (4): it would be useful to replace the phrase by "of the form ..." by "of a bilinear form in creation/annihilation operators".
  6. In Eq. (9) please define the ket-vectors $|x,s>$ and the operator $u$ between them properly. What do the indices $s$ stand for here?
  7. The expression for the many-body evolution operator $\mathcal{U}$ just before Eq. (12). is not clear to me. Please clarify the origin of $1/2$ there.
  8. Please clarify the phrase "We consider an average over separable initial states, which is equivalent to taking $\rho\propto 1$ in (1), i.e. the infinite temperature ensemble" in more details: it is not clear for a general reader.
  9. Please avoid jumping between fermionic and Pauli spin pictures and different notations like $\sigma_k$ and $X, Y, Z$. It should be also clarified for a general reader, why the fermionic and spin pictures are equivalent for OTOC and the entanglement entropy calculations.
  10. Please define the Pauli strings $\mathcal{S}$ already in Sec. 3.1.
  11. In Fig. 3, please define $\sigma(t)$ for $C_1$ and $C_2$ and show its time evolution.
  12. In Fig. 3(b), the power law is visible only at the factor of 3 in the entanglement entropy growth, therefore it is hard to verify it. Is it possible to increase the range towards smaller times in order to have more clear verification? Of course, the small system size data will not provide such an opportunity, but for L=100 it is certainly possible.
  13. In terms of Eq. (33-36) it is useful to have more cartoons of the front propagation according to the update rules within each of the parts. Please consider to add such a cartoons for more clear picture for general readers.
  14. Notations $P_2$ and $P_L$ are not defined after Eq. (39). Please define them and clarify the difference between $P_2$ and $A_k$. In the current version it is rather confusing for a general reader.
  15. If the authors would not like to rearrange the manuscript according to the item 5) of the Report, I suggest them to
  16. add the explanation similar to (33-36) to (17-19),
  17. define the Pauli strings in Sec. 3,
  18. add the explanation, why the strings $S$ contributing to OTOC must have either $X$ or $Y$ at position $r$ such that they do not commute with $Z_r$.
  19. provide the analytical arguments (70) behind (32), but not only the numerical evidence.
  20. Please correct misprints, e.g.,
  21. Page 7, second paragraph of Sec. 3.1: "for operator an" -> "for an operator"
  22. Page 20, first paragraph: "guven" -> "given" and others.

---

## Round 1 · Referee Report · Anonymous (Referee 3) · 2022-9-8

Strengths

  • It contains advances in analytical calculations

Weaknesses

  • The presentation is not clear. Especially, the notations are hard to follow.
  • The results in the paper are not very surprising.

Report

In this paper, the authors carry out a detailed analysis of the operator spreading in free fermion systems with temporal noise and observe that the operator front spread diffusively. They first consider a random free fermion circuit with only parity conservation (Z2 symmetry) and noise in both spatial and temporal directions. By studying the operator spreading in the Pauli basis, they demonstrate the diffusion in operator weight. Furthermore, the author takes an alternative approach to study the operator spreading in the single-particle picture. They consider three cases: (i)Z2 symmetric, spatial-temporal noise; (ii) Z2 symmetric, temporal noise; (iii) U1 symmetric (particle number conserved) spatial-temporal noise. They carry out extensive analytic and numeric calculations to demonstrate the diffusive operator spreading.

This is a solid work and to the best of my knowledge contains technical advances in analytic calculations. However, I think the result that operator spreading shows diffusive behavior is not very surprising. The 1D free fermion circuit describes a particle hopping randomly on a 1D lattice. In the case that scattering amplitudes from different paths coherently interfere, it leads to Anderson localization. In general, if one breaks the time-translation symmetry and thus removes this coherence, which is the case studied in this paper, one would expect diffusion from the intuition of classical random walk.

To me, the most interesting result in the paper, which should be the main message, is that OTOC is governed by 2D diffusion, while TOC is determined by 1D diffusion.

Besides, I think the paper would be much easier to understand if the authors could simplify their notations.

I will consider recommending it for publication in SciPost if the authors can restructure the paper around the new main message and simplify the notation. In the following, I list some detailed questions.

-(Section 4.2.1.) Is the author able to provide an analytic derivation for Eq.(32)? If not, the author should explain why the analytic calculation is hard and one must resort to numerics.

  • (Page 4) The sentence "Furthermore, we know that the fermionic operators A(t) obey $\partial_tA(t) = −iHA(t)$" needs to be changed. I believe this is a definition, and it is different from the usual Heisenberg evolution for operators. Saying "we know that" is very confusing.

-(Page 11) I believe the sentence "Considering (22) to (26),..." means: combining equations (22-26), one can derive the following result. May the author change the wording to make it more clear?

-What does the symbol $\vee$ and $\bigvee$ in Eq(43)-(45) mean? I wonder if the author can clarify it in the text.

  • In Eq(78) and (100), do we need a summation over $\alpha$ and $\beta$?

  • Does Eq(102) work for the case $N \neq 4$? If yes, what does the bar notation mean in this case ($N \neq 4$)?

  • Fig.10 is very confusing. For example, I do not understand the legend of panel (a). What does $\bar{C_1(r,t)} = \sigma(t)$ mean?

Requested changes

-The caption of Fig.4 needs changes. After reading the main text, I understand panels a and b represent two random circuit realizations. But one needs to explain this in the caption.

---

## Editorial Decision

awaiting_resubmission